# Scalable and Stable Estimation of Amari $\alpha$-Divergence Using Random Fourier Features

Jiaolong Wang [1]   Fode Zhang [1]   Lingrui Wang [1]

## Abstract

Reliable estimation of Amari $\alpha$-divergences underpins variational inference, yet unconstrained neural critics are notoriously prone to instability. We propose a scalable estimator by constraining the critic to a Reproducing Kernel Hilbert Space (RKHS) ball and approximating the kernel via band-limited Random Fourier Features (RFF). This formulation yields a linear-time objective amenable to mini-batch stochastic optimization while avoiding the cubic complexity of Gram-matrix methods. We present a unified analysis based on a four-term error decomposition—comprising RKHS approximation, feature discretization, statistical deviation, and optimization residual. Under a spectral source condition, we derive non-asymptotic bounds establishing that the RKHS approximation bias scales as $\mathcal{O}(R^{-\gamma})$, the RFF discretization error as $\mathcal{O}(RD^{-1/2})$, and the statistical error as $\mathcal{O}(Rn^{-1/2})$. We further show that statistical non-degeneracy induces intrinsic local curvature, enabling our proposed Armijo-SGD to achieve local linear convergence. Empirical evaluations demonstrate that the RFF-RKHS estimator outperforms varying-representation baselines in stability, and applying this spectral regularization to GAN critics significantly enhances the capture of high-frequency data components.

## 1. Introduction

Estimating divergences between probability distributions is central to machine learning, statistics, and information theory, with applications in density estimation (Kennedy et al.,

2023), variational inference (Daudel et al., 2023b), mutual information estimation (Moon et al., 2021), and generative modeling (Chen et al., 2022). Beyond the Kullback–Leibler (KL) divergence, the Amari $\alpha$-divergence family forms the unique intersection of Csiszár $\phi$-divergences and Bregman divergences, combining robustness properties with an information-geometric structure that connects learning objectives across seemingly different formulations (Amari, 2009).

The Amari family is particularly attractive because it is both information-geometric and operationally flexible. It inherits information monotonicity together with a dual-flat geometric structure, while its parameter $\alpha$ continuously changes the emphasis between mode-seeking and mass-covering behavior. This flexibility is useful in variational inference and generative modeling, where different downstream tasks may require different trade-offs between sharpness, robustness, and coverage.

Accurate divergence estimation at modern scales remains challenging. Classical plug-in estimators can suffer severe bias–variance amplification in high dimensions (Zhang et al., 2022). Variational estimators avoid explicit density estimation and naturally support mini-batch optimization, but their practical performance often hinges on two factors that existing theory only partially addresses. First, the variational optimum depends on the expressiveness of the critic class, so restricting to a misspecified function family can introduce a non-negligible approximation bias even when $n$ is large. Second, the estimator inherits optimization error: the divergence estimate is computed from an empirical maximizer, and imperfect optimization can translate directly into estimation error, especially when the objective is weakly curved or highly nonconvex. Unconstrained neural critics exacerbate both issues, yielding high-variance objectives and unstable training, while RKHS critics offer a principled way to control functional complexity. However, exact kernel methods typically rely on Gram matrices and become prohibitively expensive in time and memory as $n$ grows (often effectively $O(n^3)$) (Zellinger et al., 2023). This creates a practical trade-off: while kernel methods offer necessary statistical guarantees, they lack the scalability inherent to neural parameterizations.

---

[1]Center of Statistical Research, School of Statistics and Data Science, Southwest University of Finance and Economics, Chengdu, China. Correspondence to: Fode Zhang <fredzh@swufe.edu.cn>.

*Proceedings of the 43rd International Conference on Machine Learning*, Seoul, South Korea. PMLR 306, 2026. Copyright 2026 by the author(s).

We resolve this tension by combining RKHS control with scalable random-feature approximation. We develop a variational estimator for the full Amari $\alpha$-family by restricting the critic to an RKHS ball and implementing the optimization with RFF, which turns an infinite-dimensional problem into a finite-dimensional one compatible with mini-batching and first-order methods. Starting from a power-type variational form, we use a smoothed surrogate to avoid nondifferentiability (notably when the exponents are below one) and then optimize over a linear RFF critic. A scale-invariant post-processing step converts the maximized surrogate into the final divergence estimate (Section 3).

Our theory follows an explicit four-part error accounting tailored to kernel–feature hybrids. We separate the total error into approximation by the RKHS class, discretization from the random-feature representation, sampling noise from finite data, and the residual due to imperfect optimization. This decomposition makes model misspecification and optimization effects transparent, and yields principled choices of the radius, frequency bandwidth, feature dimension, and smoothing level as functions of the sample size.

A second theme is optimization through geometry induced by random features. In variational divergence estimation, the critic objective can be poorly conditioned without structural constraints, and analyses that treat optimization error as an external nuisance do not explain when fast convergence should occur. We show that, once the critic is approximated by bandlimited RFF, the resulting finite-dimensional objective inherits favorable intrinsic local geometry: statistical non-degeneracy induces a local curvature reminiscent of Fisher information, while the RFF feature map provides a stable norm correspondence that transfers this functional curvature to the parameter space. This interplay yields a local PL-type landscape and enables local linear convergence guarantees for ArmijoSGD (Section 4.5), clarifying how the RFF structure improves optimization stability and speed in large-scale variational estimation.

We empirically validate both the statistical scaling and the optimization stability predicted by our theory. First, we study divergence estimation on controlled synthetic distributions across dimensions and large sample sizes (Section 6.1). Second, we demonstrate an application to generative modeling on CelebA, where we inject fixed RFF into a neural critic and enlarge the frequency window; this modification improves the critic's ability to capture high-frequency information and stabilizes training under the smoothed Amari objective with an Adam+Armijo optimizer (Section 6.2). We further discuss the relation to WGAN and MMD-GAN in Appendix A.

**Contributions.** (i) We propose a scalable variational framework for the Amari $\alpha$-divergence family. By approx-

imating RKHS critics with bandlimited RFF, we bypass the cubic complexity of exact kernel methods without sacrificing their rigorous functional regularization. (ii) We provide a unified four-way error decomposition (RKHS approximation, feature discretization, sampling, optimization) and derive non-asymptotic convergence guarantees under a spectral source condition, including explicit schedules coupling $(R, W, D, \varepsilon, \lambda)$. (iii) We show that the optimization error admits sharp control because the RKHS constraint and bandlimited RFF induce favorable intrinsic local geometry, and we establish local linear convergence guarantees for ArmijoSGD without relying on large regularization. (iv) We validate the theory and demonstrate practical utility through large-scale synthetic divergence estimation and GAN experiments showing that injecting RFF and enlarging the frequency window improves high-frequency capture in neural critics.

**Conflict of Interest Disclosure.**  The authors declare no financial conflicts of interest related to this work.

## 2. Background

### 2.1. Amari $\alpha$-Divergence

We use the information–geometric Amari $\alpha$-divergence as the target discrepancy. For distributions $P, Q$ on $\mathcal{X}$ with densities $p, q$, a parameter $\alpha \in \mathbb{R} \setminus \{0, 1\}$, mutual absolute continuity, and assuming the integral exists, we adopt the normalization

$$D_A^{(\alpha)}(P\|Q) = \frac{1}{\alpha(1-\alpha)}\Big(1 - \int_{\mathcal{X}} p(x)^\alpha\, q(x)^{1-\alpha}\, dx\Big). \tag{1}$$

This divergence is nonnegative and satisfies the duality $D_A^{(\alpha)}(P\|Q) = D_A^{(1-\alpha)}(Q\|P)$. It is convex in each argument for fixed $\alpha$, and it recovers standard special cases: as $\alpha \to 1$ it converges to $D_{\mathrm{KL}}(P\|Q)$, as $\alpha \to 0$ it converges to $D_{\mathrm{KL}}(Q\|P)$, and $\alpha = \frac{1}{2}$ is proportional to the squared Hellinger distance. In this paper we work exclusively with the parameterization (1), which matches the variational objectives and the notation used in our error decomposition. For completeness, an alternative parameterization and its equivalence, as well as the associated information-geometric (dually flat) structure, are summarized in Appendix B.4 and Appendix B.5 (see also (Amari, 2009)).

### 2.2. RFF and Kernel Methods

Let $k : \mathcal{X} \times \mathcal{X} \to \mathbb{R}$ be a positive definite kernel with RKHS $(\mathcal{H}_k, \langle \cdot, \cdot \rangle_k)$. The reproducing property states that

$$f(x) = \langle f, k(x, \cdot) \rangle_k, \qquad \forall f \in \mathcal{H}_k.$$

Kernel methods often enforce capacity control by restricting the critic to an RKHS ball $\{f \in \mathcal{H}_k : \|f\|_k \le R\}$,

which yields stability and a direct handle on generalization via norm-based complexity control. Exact kernel implementations, however, require forming and factorizing an $n \times n$ Gram matrix, which is memory-intensive and typically scales superlinearly in $n$.

To obtain a scalable surrogate while retaining RKHS-style regularization, we use RFF for *shift-invariant* kernels $k(x, z) = \kappa(x - z)$ on $\mathbb{R}^d$. By Bochner's theorem, $\kappa$ is the Fourier transform of a nonnegative spectral density $p(\omega)$, hence

$$k(x, z) = \mathbb{E}_{\omega \sim p}\left[e^{i\omega^\top (x-z)}\right]. \tag{2}$$

Sampling $\omega_1, \ldots, \omega_D \overset{\text{i.i.d.}}{\sim} p$ and phases $b_1, \ldots, b_D \overset{\text{i.i.d.}}{\sim}$ Unif$[0, 2\pi]$, we define the explicit feature map

$$\psi(x) = \sqrt{\frac{2}{D}}\big(\cos(\omega_1^\top x + b_1), \ldots, \cos(\omega_D^\top x + b_D)\big)^\top \in \mathbb{R}^D, \tag{3}$$

so that $\mathbb{E}\big[\psi(x)^\top \psi(z)\big] = k(x, z)$. This replaces infinite-dimensional optimization in $\mathcal{H}_k$ by a finite-dimensional linear class $V_D = \{x \mapsto w^\top \psi(x) : w \in \mathbb{R}^D\}$, enabling minibatch training with per-iteration cost linear in $D$. Moreover, $\|w\|_2$ acts as a natural proxy for the RKHS norm, allowing us to preserve norm-based regularization in the RFF parameterization. A fuller Mercer-operator viewpoint and the precise relationship between spectral properties and RKHS approximation are deferred to Appendix B.1 (see also (Bach, 2017)).

## 3. Method

We estimate the Amari $\alpha$-divergence by maximizing a power-type variational objective over a regularized critic, implemented efficiently with RFF. The final divergence estimate uses a scale-invariant post-processing step that removes the effect of regularization from the reported value. Define the $\alpha$-affinity

$$I_\alpha(P, Q) := \int_{\mathcal{X}} p(x)^\alpha q(x)^{1-\alpha} dx,$$

so that

$$D_\alpha^{(A)}(P\|Q) = \big(1 - I_\alpha(P, Q)\big)/(\alpha(1 - \alpha)).$$

Fix exponents $0 < a < b$ and set $\alpha = b/(b-a)$. For a critic $f : \mathcal{X} \to \mathbb{R}$, consider

$$J(f) := \mathbb{E}_{x \sim P}\big[|f(x)|^a\big] - \mathbb{E}_{y \sim Q}\big[|f(y)|^b\big].$$

For a fixed $\alpha$, the pair $(a, b)$ is not unique. In practice we choose the smallest admissible exponents compatible with the desired $\alpha$, since larger powers increase the sensitivity of $\phi'_{p,\varepsilon}$ to large critic values and hence inflate gradient variance.

Maximizing over a scalar amplitude $t > 0$ yields a closed-form envelope and a constant $C_{a,b}$:

$$\sup_{f \neq 0} \sup_{t > 0} J(tf) = C_{a,b} I_\alpha(P, Q)^{1/\alpha}, \tag{4}$$

and hence

$$I_\alpha(P, Q) = \left(\tfrac{1}{C_{a,b}} \sup_{f \neq 0} \sup_{t > 0} J(tf)\right)^\alpha. \tag{5}$$

We defer the scalar envelope derivation (including $t^\star(f)$ and the explicit form of $C_{a,b}$) to Appendix B.2.

**Regularization and RFF implementation.** Direct empirical maximization of $J(f)$ is ill-posed without capacity control. We therefore constrain the critic to an RKHS $(\mathcal{H}_k, \|\cdot\|_{\mathcal{H}_k})$ with bounded kernel $k$ on compact $\mathcal{X}$, and optimize the regularized empirical objective

$$\max_{f \in \mathcal{H}_k} \widehat{J}_n(f) := \frac{1}{m}\sum_{i=1}^m |f(x_i)|^a - \frac{1}{n}\sum_{j=1}^n |f(y_j)|^b \\ - \frac{\lambda}{2}\|f\|_{\mathcal{H}_k}^2. \tag{6}$$

where $\lambda > 0$ stabilizes learning of the critic direction.

While our theory assumes Ivanov constraints ($\|f\|_{\mathcal{H}_k} \leq R$), we optimize the equivalent Tikhonov form (Page & Grünewälder, 2019). With RFF discretization $f(x) = w^\top \psi(x)$, we utilize $\|w\|_2$ as a proxy for $\|f\|_{\mathcal{H}_k}$, implementing the regularization as $\frac{\lambda}{2}\|w\|_2^2$ in the finite-dimensional objective. This turns (6) into a finite-dimensional problem

$$\widehat{J}_n(w) := 1/m\sum_{i=1}^m \big|w^\top \psi(x_i)\big|^a - 1/n\sum_{j=1}^n \big|w^\top \psi(y_j)\big|^b \\ - \lambda/2\,\|w\|_2^2. \tag{7}$$

To avoid nondifferentiability at $0$ and to stabilize optimization when $a$ or $b$ is below $1$, we optimize a smoothed surrogate obtained by replacing $|u|^p$ with $\phi_{p,\varepsilon}(u) := (u^2 + \varepsilon^2)^{p/2}$:

$$\widehat{J}_{n,\varepsilon}(w) := \frac{1}{m}\sum_{i=1}^m \phi_{a,\varepsilon}\big(w^\top \psi(x_i)\big) - \frac{1}{n}\sum_{j=1}^n \phi_{b,\varepsilon}\big(w^\top \psi(y_j)\big). \tag{8}$$

Its gradient is

$$\nabla \widehat{J}_{n,\varepsilon}(w) = \frac{1}{m}\sum_{i=1}^m \alpha_i(w)\psi(x_i) - \frac{1}{n}\sum_{j=1}^n \beta_j(w)\psi(y_j),$$

where $\quad \alpha_i(w) := a\big[(w^\top \psi(x_i))^2 + \varepsilon^2\big]^{\frac{a}{2}-1}(w^\top \psi(x_i)),$

$$\beta_j(w) := b\big[(w^\top \psi(y_j))^2 + \varepsilon^2\big]^{\frac{b}{2}-1}(w^\top \psi(y_j)). \tag{9}$$

We optimize (8) by minibatch stochastic gradient ascent. Even when the powers are differentiable, smoothing remains useful for optimization: for $1 \le p < 2$, the unsmoothed second derivative may be unbounded near zero, while for larger powers gradients can be weak around initialization. The parameter $\varepsilon$ bounds the local curvature and provides a small but stable numerical floor, while its bias is controlled explicitly in Section 4. The smoothing mechanism and basic approximation bounds are summarized in Appendix B.3.

**Scale-invariant post-processing.** The scalar post-processing is not a second critic optimization. The population variational identity is scale-invariant because it maximizes over both the direction $f$ and the amplitude $t > 0$. In contrast, the practical smoothed and regularized empirical objective used to learn $\hat{w}$ is not scale-invariant: the magnitude of $\hat{w}$ depends on $\lambda$, $\varepsilon$, and the optimization trajectory. We therefore use the first stage to learn a stable critic direction and the second stage to recover the optimal amplitude analytically. Let $\hat{w}$ denote the optimizer output. Since $\widehat{J}_{n,\varepsilon}(\hat{w})$ depends on $\lambda$ and the scale of $\hat{w}$, we remove regularization bias by re-optimizing only a scalar amplitude $t > 0$ while fixing the direction $\hat{w}$:

$$\hat{A} := \frac{1}{m}\sum_{i=1}^{m}\left|\hat{w}^\top\psi(x_i)\right|^a, \hat{B} := \frac{1}{n}\sum_{j=1}^{n}\left|\hat{w}^\top\psi(y_j)\right|^b,$$

$$\hat{t} := \left(\frac{a\,\hat{A}}{b\,\hat{B}}\right)^{\frac{1}{b-a}}. \tag{10}$$

This yields the empirical envelope

$$\widehat{\mathcal{V}}_{a,b} := \sup_{t>0}\left\{\frac{1}{m}\sum_{i=1}^{m}\left|t\,\hat{w}^\top\psi(x_i)\right|^a - \frac{1}{n}\sum_{j=1}^{n}\left|t\,\hat{w}^\top\psi(y_j)\right|^b\right\}$$

$$= C_{a,b}\,\hat{A}^{\frac{b}{b-a}}\,\hat{B}^{-\frac{a}{b-a}} \tag{11}$$

and the final plug-in estimator becomes

$$\widehat{D_\alpha^{(A)}}(P\|Q) = \frac{1}{\alpha(1-\alpha)}\left(1 - \left(\frac{\widehat{\mathcal{V}}_{a,b}}{C_{a,b}}\right)^\alpha\right), \quad \alpha = \frac{b}{b-a}. \tag{12}$$

## 4. Theoretical Analysis

To conduct a comprehensive error analysis, we meticulously decomposed the total error into four components. We provided explicit bounds and derived non-asymptotic convergence rates under appropriate assumptions.

### 4.1. Error Telescoping with a Proxy Objective

Let $P$ and $Q$ be probability distributions on a compact domain $\mathcal{M} \subset \mathbb{R}^d$. Fix exponents $a, b > 0$ (typically $a < b$) and consider the population objective

$$J(f) = \mathbb{E}_{x\sim P}\left[|f(x)|^a\right] - \mathbb{E}_{y\sim Q}\left[|f(y)|^b\right], \qquad f : \mathcal{M} \to \mathbb{R},$$

under the boundedness constraint $\|f\|_\infty \le B$. Let $\mathcal{H}_k$ be the RKHS of a bounded kernel $k$ on $\mathcal{M}$, with $S_k := \sup_{x\in\mathcal{M}} k(x,x) < \infty$, so that $\|f\|_\infty \le \sqrt{S_k}\|f\|_{\mathcal{H}_k}$ for all $f \in \mathcal{H}_k$. Let $\psi : \mathcal{M} \to \mathbb{R}^D$ be a random Fourier feature map and define the finite-dimensional class

$$V_D := \left\{x \mapsto w^\top\psi(x) : w \in \mathbb{R}^D\right\} \subset \mathcal{H}_k. \tag{13}$$

Given samples $\{x_i\}_{i=1}^{m} \sim P$ and $\{y_j\}_{j=1}^{n} \sim Q$, write the empirical objective

$$\widehat{J}_n(f) = \frac{1}{m}\sum_{i=1}^{m}|f(x_i)|^a - \frac{1}{n}\sum_{j=1}^{n}|f(y_j)|^b.$$

**A smooth proxy objective.** To avoid the nonsmooth behavior of power functions near the origin (which prevents global Lipschitz/smooth arguments when $a < 1$ or $b < 1$), we introduce the smoothed map

$$\phi_{p,\varepsilon}(t) := (t^2 + \varepsilon^2)^{p/2},$$
$$J_\varepsilon(f) := \mathbb{E}_P[\phi_{a,\varepsilon}(f)] - \mathbb{E}_Q[\phi_{b,\varepsilon}(f)].$$

with empirical counterpart $\widehat{J}_{n,\varepsilon}$. The proxy uniformly approximates the original objective:

$$\left|J(f) - J_\varepsilon(f)\right| \le \varepsilon^a + \varepsilon^b, \qquad \forall f. \tag{14}$$

**Reference functions.** Let $f^\star \in \arg\max_{\|f\|_\infty\le B} J(f)$ be a population maximizer. Fix an RKHS radius $R > 0$ and define the truncated RKHS ball

$$\mathcal{B}_R := \{g \in \mathcal{H}_k : \|g\|_{\mathcal{H}_k} \le R, \|g\|_\infty \le B\}.$$

We introduce the bandlimited proxy and its finite-feature approximation

$$g_R \in \mathcal{B}_R,$$
$$g_{D,R}^\dagger \in \arg\min_{v\in V_D\cap\mathcal{B}_R}\|g_R - v\|_\infty.$$

Here $g_R$ denotes the bandlimited RKHS proxy constructed in Theorem 4.3; using this explicit proxy makes the feature-discretization step compatible with the bandlimited RFF approximation lemma. For the proxy objective, let

$$\widehat{g}_{D,R}^\star \in \arg\max_{v\in V_D\cap\mathcal{B}_R}\widehat{J}_{n,\varepsilon}(v), \qquad \widehat{g}_{\varepsilon,T} \in V_D\cap\mathcal{B}_R$$

denote, respectively, an empirical maximizer and the output of an optimization algorithm after $T$ iterations. Adding and subtracting the proxy and reference functions yields the

telescoping identity

$$
\begin{aligned}
J(f^\star) &- \widehat{J}_{n,\varepsilon}(\widehat{g}_{\varepsilon,T}) \\
&= \underbrace{\left[ J(f^\star) - J_\varepsilon(f^\star) \right]}_{\mathcal{E}_{\text{proxy}}} + \underbrace{\left[ J_\varepsilon(f^\star) - J_\varepsilon(g_R) \right]}_{\mathcal{E}_{\text{RKHS}}} \\
&+ \underbrace{\left[ J_\varepsilon(g_R) - J_\varepsilon(g_{D,R}^\dagger) \right]}_{\mathcal{E}_{\text{feat}}} + \underbrace{\left[ J_\varepsilon(g_{D,R}^\dagger) - \widehat{J}_{n,\varepsilon}(g_{D,R}^\dagger) \right]}_{\mathcal{E}_{\text{stat}}} \\
&+ \underbrace{\left[ \widehat{J}_{n,\varepsilon}(g_{D,R}^\dagger) - \widehat{J}_{n,\varepsilon}(\widehat{g}_{D,R}^\star) \right]}_{\mathcal{E}_{\text{mis}}} + \underbrace{\left[ \widehat{J}_{n,\varepsilon}(\widehat{g}_{D,R}^\star) - \widehat{J}_{n,\varepsilon}(\widehat{g}_{\varepsilon,T}) \right]}_{\mathcal{E}_{\text{opt}}}
\end{aligned}
$$
$$(15)$$

For readability, we use semantic labels rather than the visual labels (A)–(D). Here $\mathcal{E}_{\text{proxy}}$ is the explicit proxy bias, bounded by (14); $\mathcal{E}_{\text{RKHS}}$ and $\mathcal{E}_{\text{feat}}$ are the two approximation biases induced by restricting the search space from $\mathcal{M}_b$ to $\mathcal{B}_R$ and then to $V_D \cap \mathcal{B}_R$; $\mathcal{E}_{\text{stat}}$ is the sampling error of the proxy objective at a fixed reference; $\mathcal{E}_{\text{mis}} \leq 0$ is the nonpositive empirical mismatch between the reference and the empirical maximizer; and $\mathcal{E}_{\text{opt}} \geq 0$ is the algorithmic suboptimality after $T$ iterations.

Define the uniform proxy generalization gap

$$
\mathcal{E}_{\text{gen}} := \sup_{g \in V_D \cap \mathcal{B}_R} \left| J_\varepsilon(g) - \widehat{J}_{n,\varepsilon}(g) \right|.
$$

A standard argument (stated and proved in Appendix C.2) shows that

$$
\mathcal{E}_{\text{stat}} + \mathcal{E}_{\text{mis}} \ \leq \ 3\,\mathcal{E}_{\text{gen}}. \tag{16}
$$

Thus, after controlling $\mathcal{E}_{\text{gen}}$, the only remaining algorithm-dependent term is the optimization residual $\mathcal{E}_{\text{opt}}$, while conclusions transfer back to the original objective $J$ through the explicit bridge (14). The following subsections bound $\mathcal{E}_{\text{RKHS}}$, $\mathcal{E}_{\text{feat}}$, $3\mathcal{E}_{\text{gen}}$, and $\mathcal{E}_{\text{opt}}$ in turn.

### 4.2. Function Approximation in RKHS

We bound the RKHS approximation term $\mathcal{E}_{\text{RKHS}}$ in (15) by quantifying how well the population maximizer $f^\star$ can be approximated by an RKHS ball of radius $R$. For shift-invariant kernels, it is convenient to express regularity and truncation directly in the Fourier domain (via Bochner's theorem), which aligns with the bandlimited search directions used by our optimization algorithm.

**Assumption 4.1** (Bandlimit and spectral window). There exists $W_0 > 0$ such that all search directions $h$ produced by the optimization algorithm belong to

$$
\mathcal{B}_W(\Omega) := \left\{ h \in L^2(\Omega) : \widehat{h}(\omega) = 0 \text{ for all } |\omega| > W \right\}
$$

for some $W \geq W_0$. Moreover, for each $W \geq W_0$ there exist finite constants $m_k(W), M_k(W)$ such that

$$
0 < m_k(W) \leq \hat{k}(\omega) \leq M_k(W) < \infty, \qquad \forall\,|\omega| \leq W,
$$

where $\hat{k}$ denotes the spectral density of the shift-invariant kernel $k$.

**Assumption 4.2** (Fourier-domain source condition). There exists $0 < r < \frac{1}{2}$ and $\xi \in L^2(\mathbb{R}^d)$ such that

$$
\widehat{f^\star}(\omega) = \hat{k}(\omega)^r\,\widehat{\xi}(\omega), \qquad \|\xi\|_{L^2(\mathbb{R}^d)} \leq M_\xi.
$$

Under these assumptions, truncating the Fourier transform of $f^\star$ yields an RKHS element with controlled norm and an explicit $L^2$ approximation rate.

**Theorem 4.3** (RKHS approximation via bandlimited truncation). *Let $k$ be shift-invariant on $\mathbb{R}^d$ with radially nonincreasing $\hat{k}$ and suppose Assumptions 4.1 and 4.2 hold with $0 < r < \frac{1}{2}$. Then for any sufficiently large $R > 0$ there exists a bandlimited $g_R \in \mathcal{H}_k$ with*

$$
\widehat{g_R}(\omega) = 0 \text{ for all } |\omega| > W(R), \qquad \|g_R\|_{\mathcal{H}_k} \leq R,
$$

*such that*

$$
\|f^\star - g_R\|_{L^2(\mathbb{R}^d)} \ \leq \ C_1\,R^{-\frac{2r}{1-2r}}. \tag{17}
$$

The proof is deferred to Appendix D.1. Next, by assuming $f^\star \in H^s(\mathcal{M})$ for $s > d/2$ (which ensures the approximation error remains bounded in Sobolev norm), we apply interpolation (Appendix D.2) to upgrade the $L^2$ rate to a uniform bound:

$$
\|f^\star - g_R\|_\infty \ \leq \ C_\infty\,R^{-\gamma}, \qquad \gamma := \frac{2r}{1-2r}\left(1 - \frac{d}{2s}\right). \tag{18}
$$

Finally, recall $\mathcal{E}_{\text{RKHS}} = J_\varepsilon(f^\star) - J_\varepsilon(g_R)$, where $g_R$ is the bandlimited RKHS proxy from Theorem 4.3. Since $J_\varepsilon$ is Lipschitz on $\{f : \|f\|_\infty \leq B\}$, there exists $L_{\varepsilon,B} < \infty$ such that

$$
\mathcal{E}_{\text{RKHS}} \ \leq \ L_{\varepsilon,B}\,\|f^\star - g_R\|_\infty \ \lesssim \ R^{-\gamma} \tag{19}
$$

where the last step uses (18). Equation (19) is the bound used in the master decomposition (15).

### 4.3. Feature Approximation with RFF

We bound the *feature approximation term* $\mathcal{E}_{\text{feat}}$ in (15). Fix the bandlimited proxy $g_R \in \mathcal{H}_k$ from Theorem 4.3, with bandwidth $W(R)$ and $\|g_R\|_{\mathcal{H}_k} \leq R$, and define

$$
g_{D,R}^\dagger \in \arg\min_{v \in V_D \cap \mathcal{B}_R} \|g_R - v\|_\infty,
$$
$$
\mathcal{E}_{\text{feat}} := J_\varepsilon(g_R) - J_\varepsilon(g_{D,R}^\dagger).
$$

Since $J_\varepsilon$ is Lipschitz on $\{f : \|f\|_\infty \leq B\}$, there exists $L_{\varepsilon,B} < \infty$ such that

$$
|\mathcal{E}_{\text{feat}}| \ \leq \ L_{\varepsilon,B}\,\|g_R - g_{D,R}^\dagger\|_\infty. \tag{20}
$$

To control $\|g_R - g_{D,R}^\dagger\|_\infty$ we use an improved uniform RFF approximation bound that exploits the *bandlimit* of the target (full statement and proof in Appendix E). Concretely, under Assumption 4.1, if the random features are sampled from the truncated spectral measure associated with the window $\|\omega\| \leq W(R)$, then with probability at least $1 - \delta$ there exists $v_{D,W(R)} \in V_D$ such that

$$\|g_R - v_{D,W(R)}\|_\infty \leq C_{\mathrm{RFF}} R\Big(W(R)\sqrt{\frac{d}{D}} + \sqrt{\frac{\log(1/\delta)}{D}}\Big),$$

where $C_{\mathrm{RFF}}$ depends only on $k$ and $\mathrm{diam}(\mathcal{M})$. By the definition of $g_{D,R}^\dagger$, $\|g_R - g_{D,R}^\dagger\|_\infty \leq \|g_R - v_{D,W(R)}\|_\infty$, and substituting into (20) yields the final bound:

$$|\mathcal{E}_{\mathrm{feat}}| \leq L_{\varepsilon,B}\, C_{\mathrm{RFF}}\, R\big(W(R)\sqrt{d/D} + \sqrt{\log(1/\delta)/D}\big), \tag{21}$$

with probability at least $1 - \delta$.

**Bias–variance coupling through the spectral window.**
The window $W(R)$ is the key coupling parameter across the approximation stages. A larger window lowers the intrinsic RKHS approximation bias (Term $\mathcal{E}_{\mathrm{RKHS}}$) by allowing $g_R$ to retain higher-frequency content of $f^\star$ under the source condition. However, the same $W(R)$ enlarges the effective complexity of the bandlimited proxy as seen by RFF, inflating the discretization fluctuation in (21) through the factor $W(R)\sqrt{d/D}$. Consequently, maintaining a target accuracy requires increasing $D$ as $W(R)$ grows, which makes explicit how the feature budget must scale with the RKHS radius (and hence with the bias level) to keep the overall estimator consistent.

### 4.4. Statistical Error Analysis

We bound the statistical component in (15),

$$\mathcal{E}_{\mathrm{stat}} = J_\varepsilon(g_{D,R}^\dagger) - \widehat{J}_{n,\varepsilon}(g_{D,R}^\dagger),$$

via the uniform proxy deviation

$$\mathcal{E}_{\mathrm{gen}} := \sup_{g \in V_D} \big|J_\varepsilon(g) - \widehat{J}_{n,\varepsilon}(g)\big|, \qquad |\mathcal{E}_{\mathrm{stat}}| \leq \mathcal{E}_{\mathrm{gen}}.$$

Rather than imposing an external Sobolev smoothness assumption, we exploit the bandlimited structure from Assumption 4.1. Bandlimited RKHS functions are real-analytic on compact $\mathcal{M}$, which implies a Sobolev-type entropy bound for bandlimited RKHS balls; see Lemma F.1 in Appendix F. This entropy control enables a Dudley/Talagrand chaining bound with a *parametric* $N^{-1/2}$ rate once we choose an effective smoothness index $h(W) > d$.

**Theorem 4.4** (Uniform proxy deviation under bandlimited entropy). *Assume $V_D \subset \mathcal{H}_{k,W}$ for some $W \geq W_0$ and $\|g\|_{\mathcal{H}_k} \leq R$ for all $g \in V_D$. Let $S_k := \sup_{x \in \mathcal{M}} k(x,x)$ and set $B := \sqrt{S_k}\, R$ so that $\|g\|_\infty \leq B$ on $V_D$. Let*

$h(W) > d$ *and* $C_h(W)$ *be the bandlimited entropy constants from Lemma F.1. Then for any* $\delta > 0$,

$$\Pr\Big(\mathcal{E}_{\mathrm{gen}} \geq C\, A_{\varepsilon,W}\, (m^{-1/2} + n^{-1/2}) + \delta\Big)$$
$$\leq \exp\Big(-c\, \frac{m\delta^2}{M_{a,\varepsilon}^2}\Big) + \exp\Big(-c\, \frac{n\delta^2}{M_{b,\varepsilon}^2}\Big), \tag{22}$$

*where $c, C > 0$ are universal constants and $A_{\varepsilon,W}$ depends on $(\varepsilon, R, W)$ only through $A_{\varepsilon,W} = C_h(W)\, R\, \sqrt{S_k}\, L_\varepsilon$.*

In (22), $L_\varepsilon := \max\{L_{a,\varepsilon}, L_{b,\varepsilon}\}$ with

$$L_{p,\varepsilon} := \sup_{t \in \mathbb{R}} |\phi_{p,\varepsilon}'(t)| < \infty,$$
$$M_{p,\varepsilon} := \sup_{|t| \leq B} \phi_{p,\varepsilon}(t) = (B^2 + \varepsilon^2)^{p/2}. \tag{23}$$

See Appendix F.2 for the proof.

Let $N := m \wedge n$. Choosing $\delta$ of the same order as the leading term in (22) yields

$$\mathcal{E}_{\mathrm{gen}} = O_{\mathrm{Pr}}\big(A_{\varepsilon,W}\, N^{-1/2}\big),$$
$$|\mathcal{E}_{\mathrm{stat}}| \leq \mathcal{E}_{\mathrm{gen}} = O_{\mathrm{Pr}}\big(A_{\varepsilon,W}\, N^{-1/2}\big).$$

The window $W$ acts as a coupling parameter through $C_h(W)$: enlarging $W$ increases the statistical constant (via larger metric entropy), while it reduces approximation bias (Term $\mathcal{E}_{\mathrm{RKHS}}$) and also inflates the RFF discretization fluctuation (Term $\mathcal{E}_{\mathrm{feat}}$) through $W(R)$. This makes explicit how $W$ simultaneously governs sampling complexity, approximation bias, and feature variance.

### 4.5. Optimization Error via Intrinsic Local Geometry

We control the *empirical optimization residual* in (15),

$$\mathcal{E}_{\mathrm{opt}}(T) := \widehat{J}_{n,\varepsilon}\big(\widehat{g}_{D,R}^\star\big) - \widehat{J}_{n,\varepsilon}\big(\widehat{g}_{\varepsilon,T}\big) \geq 0, \tag{24}$$

where $\widehat{g}_{D,R}^\star$ maximizes $\widehat{J}_{n,\varepsilon}$ over the feasible class and $\widehat{g}_{\varepsilon,T}$ is the algorithm iterate. The key point is that the *fast local rate* is driven by *intrinsic curvature induced by statistical non-degeneracy* (a Fisher-information-like term), while the regularizer $\lambda$ contributes only additively.

**Regularized proxy and ArmijoSGD.** On the random-feature class $V_D = \{f_w(\cdot) = w^\top \psi(\cdot) : w \in \mathbb{R}^D\}$, define

$$\widehat{J}_{n,\varepsilon}(f_w) = \frac{1}{m}\sum_{i=1}^m \phi_{a,\varepsilon}\big(w^\top \psi(x_i)\big) - \frac{1}{n}\sum_{j=1}^n \phi_{b,\varepsilon}\big(w^\top \psi(y_j)\big), \tag{25}$$

and optimize over $\mathbb{B}_{R_w} := \{w \in \mathbb{R}^D : \|w\|_2 \leq R_w\}$. For stability and geometry, we work with the regularized *minimization* objective

$$\widehat{F}(w) := -\widehat{J}_{n,\varepsilon}(f_w) + \frac{\lambda}{2}\|w\|_2^2, \qquad w_\lambda^\star \in \arg\min_{w \in \mathbb{B}_{R_w}} \widehat{F}(w). \tag{26}$$

We run ArmijoSGD with stochastic gradients $g_t$ satisfying $\mathbb{E}[g_t \mid w_t] = \nabla \widehat{F}(w_t)$ and bounded additive variance; see Appendix G for the formal assumptions. Given $\eta_{\max} > 0$, $\beta \in (0,1)$, $c \in (0,1)$ and $K_{\max} \in \mathbb{N}$, ArmijoSGD chooses $\eta_t \in \{\eta_{\max}\beta^k\}_{k=0}^{K_{\max}}$ as the first step satisfying the Armijo decrease (53) (else $\eta_t = 0$), and updates by projection (54). Let $\eta_{\min} := \eta_{\max}\beta^{K_{\max}}$. Although $\widehat{F}$ is generally non-convex, statistical non-degeneracy induces a *local* strong-convexity/PL region around $w_\lambda^\star$. In Appendix G we show: (i) under mild regularity of $Q$ and the random-feature sampling, the feature covariance $\Sigma_Q = \mathbb{E}_{y \sim Q}[\psi(y)\psi(y)^\top]$ is strictly positive definite a.s.; (ii) a *non-degeneracy margin* $\kappa_{\mathrm{nd}} > 0$ (depending on $(a, b, \varepsilon)$ and the density-ratio range) lower-bounds the population data curvature by a Fisher-like form $\kappa_{\mathrm{nd}}\Sigma_Q$; and (iii) consequently, there exists a neighborhood

$$\mathcal{S}_\star := \left\{ w \in \mathbb{B}_{R_w} : \|w - w_\lambda^\star\|_2 \le r_\star \right\}$$

on which $\widehat{F}$ satisfies a local PL inequality with constant $\mu_{\mathrm{loc}} > 0$ (precise definitions and proofs are in Appendix G). This separates the roles of curvature sources: $\mu_{\mathrm{loc}}$ is governed primarily by intrinsic data geometry, while $\lambda$ enters only through the additive $\lambda I_D$ term in the Hessian.

**Theorem 4.5** (Local linear convergence of ArmijoSGD (stopped process)). *Let $\tau := \inf\{t \ge 0 : w_t \notin \mathcal{S}_\star\}$ and*

$$\Delta_F(t) := \widehat{F}(w_{t \wedge \tau}) - \widehat{F}(w_\lambda^\star) \ge 0.$$

*Assume $\widehat{F}$ is $L$-smooth on $\mathbb{B}_{R_w}$ and satisfies a local PL inequality with constant $\mu_{\mathrm{loc}}$ on $\mathcal{S}_\star$. Run ArmijoSGD with $\eta_{\max} \le (2L)^{-1}$, and suppose that for all $t < \tau$ the backtracking accepts a non-null step (so $\eta_t \ge \eta_{\min}$). Then for all $t \ge 0$,*

$$\mathbb{E}\big[\Delta_F(t)\big] \le (1 - \eta_{\min}\mu_{\mathrm{loc}})^t \Delta_F(0) + \frac{L\eta_{\max}^2}{2\,\eta_{\min}\mu_{\mathrm{loc}}}\,\sigma^2. \tag{27}$$

**Bridge to $\mathcal{E}_{\mathrm{opt}}(T)$.** Let $w^\star$ maximize the unregularized $\widehat{J}_{n,\varepsilon}$ on $\mathbb{B}_{R_w}$. Using $\widehat{J}_{n,\varepsilon} = -\widehat{F} + \frac{\lambda}{2}\|\cdot\|_2^2$, we decompose the error as $\mathcal{E}_{\mathrm{opt}}(T) = \widehat{F}(w_T) - \widehat{F}(w^\star) + \frac{\lambda}{2}(\|w^\star\|_2^2 - \|w_T\|_2^2)$. Since $\widehat{F}(w^\star) \ge \min \widehat{F} = \widehat{F}(w_\lambda^\star)$ and $\|w^\star\|_2 \le R_w$, we bound this by $(\widehat{F}(w_T) - \widehat{F}(w_\lambda^\star)) + \frac{\lambda}{2}R_w^2$. Applying the local geometry bound yields:

$$\mathcal{E}_{\mathrm{opt}}(T) \le C_{\mathrm{geo}}\sqrt{\frac{\widehat{F}(w_{T \wedge \tau}) - \widehat{F}(w_\lambda^\star)}{\mu_{\mathrm{loc}}}} + \frac{\lambda}{2}R_w^2. \tag{28}$$

## 5. Algorithm and Design of the Improved Estimator

This section turns the four-term decomposition in Section 4 into an implementable estimator. We keep the presentation short and focus on the resulting *schedules* for

$(\varepsilon, R, W, D, \lambda)$. The full pseudocode and the derivations behind these choices appear in Appendix J (Algorithm 1).

Our estimator follows three principles established in Section 4. First, we optimize the smoothed proxy objective $J_\varepsilon$ to handle the non-smooth regime $(a, b < 1)$, with proxy bias controlled by $\varepsilon$. Second, we enforce the spectral coupling between the RKHS radius $R$ and the bandlimit $W = W(R)$ to balance intrinsic approximation bias and sampling/feature variance. Third, we stabilize optimization using intrinsic local geometry (Section 4.5) rather than relying on large regularization.

We work with the bandlimited random-feature class induced by truncating the spectral measure to $\{\|\omega\| \le W\}$ and using the explicit RFF map $\psi(\cdot)$ in (3). This enforces Assumption 4.1 at the algorithmic level and yields a finite-dimensional critic $f_w(x) = w^\top \psi(x)$ with $\|w\|_2 \le R_w$.

We optimize the regularized empirical proxy objective $\widehat{F}$ defined in (26) using ArmijoSGD (see Algorithm 1 in Appendix J). The only role of $\lambda$ here is to ensure that the additive regularization term in the telescoping bridge, $\frac{\lambda}{2}R_w^2$ in (28), vanishes asymptotically. In particular, to keep regularization bias below the statistical scale, we set

$$\lambda R_w^2 = o(N^{-1/2}), \qquad N := m \wedge n, \tag{29}$$

while the local curvature governing convergence is provided by the intrinsic geometry (via $\mu_{\mathrm{loc}}$).

The feature budget must match the *bandlimited* RFF approximation rate in Section 5.3. Ignoring logarithmic factors, the condition $D \gtrsim d R^2 W(R)^2 N$ ensures that the feature discretization term is at most of order $N^{-1/2}$ and therefore does not dominate the statistical error (details in Appendix J). Finally, we set the smoothing level at the statistical scale, $\varepsilon \asymp N^{-1/2}$ so that the proxy bias bound $|\mathcal{E}_{\mathrm{proxy}}| \lesssim \varepsilon^a + \varepsilon^b$ balances the sampling noise and does not change the leading $N^{-1/2}$ rate in the bandlimited regime.

Algorithm 1 (Appendix J) has a short calibration stage that chooses $(R, W, D, \varepsilon, \lambda)$, followed by ArmijoSGD on $\widehat{F}$ with projection onto $\mathbb{B}_{R_w}$. Per iteration, evaluating a mini-batch gradient costs $O(BD)$ for batch size $B$. The Armijo backtracking multiplies this cost by at most $K_{\max}$, so the per-iteration cost remains $O(K_{\max}BD)$ with memory $O(D)$.

## 6. Experiments and Application

We validate the theoretical views and the practical stability of the smoothed Amari objective in two settings: divergence estimation on controlled synthetic distributions, and generative modeling on CelebA with a Spatial-RFF discriminator.

## 6.1. Divergence Estimation on Synthetic Data

We estimate the Amari $\alpha$-divergence $D_\alpha^{(A)}(P\|Q)$ between a truncated Gaussian $P$ and a uniform reference $Q$ supported on a compact union of hyper-rectangles $\mathcal{X} \subset \mathbb{R}^d$ (details in Appendix F). We report results for $d \in \{2, 5, 10\}$ and $n \in \{10^5, \ldots, 5 \times 10^6\}$, with the ground-truth divergence computed by high-precision Monte Carlo importance sampling (See Appendix K.1 for more details).

We compare four critics: (i) **Basic(NN)**: a standard MLP baseline; (ii) **Adaptive(NN)**: an MLP with width $k_n$ increasing with $n$; (iii) **RFF-Estimator**: a linear critic $f_w(x) = w^\top \psi(x)$ over RFF; and (iv) **RKHS(NN)**: a neural critic with explicit smoothness regularization to emulate RKHS geometry. Full architectural and hyperparameter specifications are deferred to Appendix K.1. Before presenting the quantitative benchmarks, we first verify the geometric superiority of the proposed RFF and RKHS methods over standard neural baselines by analyzing their training dynamics (see Appendix L Figure 2 for details). As shown in Figure 1, (iii) achieves the lowest mean error, effectively outperforming baseline (iv). This confirms that adhering to the spectral scaling law $D \propto R^2 W(R)^2$ allows the random feature approximation to match or exceed RKHS-regularized neural networks.

Regarding dimensionality, the RFF estimator proves robust. It maintains the lowest RMSE at $d = 2$ and $d = 5$, and unlike the divergent (i), remains stable and competitive at $d = 10$. Finally, while unconstrained baselines (i) and (ii) exhibit instability in low-sample regimes, the RFF estimator maintains a consistent error profile. This stability indicates that a properly scaled fixed basis mitigates overfitting and smooths the optimization landscape more effectively than (i) and (ii).

## 6.2. Generative Modeling with Spatial RFF

As a controlled critic-side ablation in image generation, CNN discriminators can underemphasize high-frequency geometric cues, potentially destabilizing critic-based training under nonstandard objectives. We therefore inject fixed high-frequency positional features via Spatial RFFs, providing a geometry-sensitive inductive bias while maintaining the discriminator's expressiveness.

The CNN experiment should be interpreted as an additive spectral-capacity implementation of the linear RFF theory. A deep discriminator locally acts as a linear readout over learned features, $D(x) = w_1^\top \Phi_{\text{CNN}}(x)$. Injecting Spatial RFF augments this representation by a fixed spectral component, $w_2^\top \psi_{\text{RFF}}(p)$, which instantiates the controlled RFF subspace analyzed in our theory while preserving the CNN backbone. Here "stability" refers to critic-side optimization stability and reliable divergence estimation, rather than a

*Table 1.* **Ablation over Spatial-RFF bandwidth $\sigma$ (CelebA).** Lower is better for FID/KID/SWD.

| Model | FID↓ | KID↓ $(\times 10^{-2})$ | SWD↓ | HF Energy |
|---|---|---|---|---|
| $\sigma = 0$ | 240.47 | 22.59 | 143.92 | 24.89 |
| $\sigma = 1$ | 63.56 | 5.14 | **98.30** | 31.64 |
| $\sigma = 2$ | 30.07 | 2.39 | 127.23 | 32.09 |
| $\sigma = 5$ | **27.23** | **2.15** | 120.78 | 32.57 |

general solution to all adversarial-training pathologies. Full GAN stability also depends on generator architecture and dynamics; our experiments isolate the effect of the RFF-regularized critic while keeping the rest of the backbone fixed.

We train GANs on CelebA and use a standard upsampling generator and a CNN discriminator augmented with Spatial RFFs; to enhance the discriminator's sensitivity to high-frequency spatial details, we move beyond simple coordinate concatenation (Liu et al., 2018). Instead, we adopt Fourier Feature mappings, which have been shown to mitigate spectral bias in neural networks (Tancik et al., 2020). Let $p \in [-1, 1]^2$ denote normalized coordinates on the discriminator bottleneck grid. We form a fixed positional embedding

$$v(p) = \big[ \cos(2\pi \mathbf{B}p), \ \sin(2\pi \mathbf{B}p) \big], \qquad \mathbf{B} \sim \mathcal{N}(0, \sigma^2 I).$$

Distinct from prior works that inject positional encodings primarily in the generator (Xu et al., 2021), we concatenate $v(p)$ with the CNN bottleneck features along channels. To avoid disrupting early training, we apply *zero-initialization* on the fusion $1 \times 1$ convolution weights corresponding to the RFF channels (CNN channels follow standard initialization), so the discriminator starts as a pure CNN and gradually incorporates spatial cues. Specifications are moved to Appendix K.2.

Table 2 reports the rebuttal validation on FFHQ-256 (Karras et al., 2018) with a modern adversarial backbone. This experiment is not intended to claim a universal generative-modeling SOTA; rather, it tests whether the same critic-side spectral regularization remains beneficial when inserted into a modern adversarial backbone. Compared with the earlier bare-bones CNN ablation, the R3GAN-based variant substantially reduces the saturation artifacts observed in the baseline samples.

We train with the smoothed Amari objective

$$\mathcal{L}(D, G) = \mathbb{E}_{p_{\text{data}}}[\phi_{a,\varepsilon}(D(x))] - \mathbb{E}_{p_z}[\phi_{b,\varepsilon}(D(G(z)))]. \tag{30}$$

While Theorem 4.5 guarantees convergence for the gradient direction, in deep learning applications (Section 6.2), we employ a heuristic *Adam+Armijo* variant. This adapts the step direction using Adam's preconditioner to handle

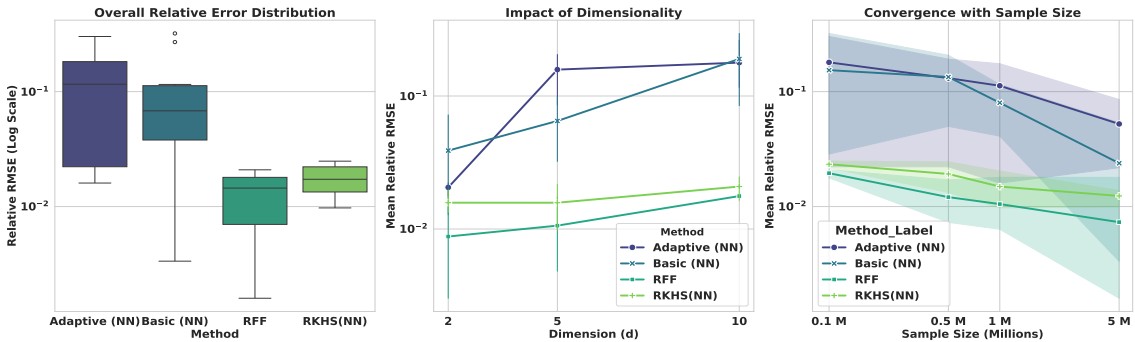

*Figure 1.* **Estimator Performance Analysis.** Comparison of estimation errors across **(Left)** relative error distribution, **(Middle)** increasing dimensionality, and **(Right)** sample size.

conditioning, while retaining the Armijo backtracking to enforce the sufficient decrease required for stability; the pseudocode and implementation details are provided in Appendix K.2 (Algorithm 2). We also apply $L_2$ regularization on discriminator weights with $\lambda = 10^{-3}$.

For the CelebA controlled ablation, we report FID, KID, SWD, and high-frequency (HF) energy on 50,000 generated samples (definitions omitted; standard in the GAN literature). Table 1 reports an ablation over the Spatial-RFF bandwidth $\sigma$. The baseline $\sigma = 0$ corresponds to removing the Spatial-RFF injection. Visual inspection of the generated samples in Figure 3 (Appendix L) aligns with the quantitative metrics. The model with $\sigma = 5$ produces coherent facial structures with sharp details, whereas severe artifacts and lack of definition characterize the baseline samples.

## 7. Conclusion

We presented a scalable framework for variational estimation of the full Amari $\alpha$-divergence family by constraining the critic to an RKHS and performing variational optimization with bandlimited Random Fourier Features. Our analysis provides a unified four-way error decomposition: RKHS approximation, feature discretization, sampling, optimization and a Fourier-domain coupling that turns these terms into concrete guidance for selecting the feature bandwidth and budget under a fixed sample size. We also showed that the RFF structure induces favorable intrinsic local geometry through feature covariance and statistical non-degeneracy, enabling stable local linear convergence guarantees for Armijo-style optimization.

Empirically, the proposed RFF estimator matches the predicted scaling behavior and improves stability relative to neural baselines on synthetic divergence estimation. In GAN training, injecting Spatial RFF into the critic and enlarging the frequency window enhances high-frequency capture, yielding better sample quality and spectral metrics. Future work includes data-adaptive spectral windows, al-

ternative feature maps beyond shift-invariant kernels, and extensions to other variational objectives where critic misspecification and optimization error dominate performance.

## Acknowledgements

This work was supported in part by the Sichuan Science and Technology Program under Grant 2024ZYD0135, in part by the National Natural Science Foundation of China under Grants 12071372 and 12201395.

## Impact Statement

This paper advances methodological and theoretical foundations for scalable variational estimation of Amari $\alpha$-divergences using RKHS critics with RFF. The primary anticipated impact is enabling more reliable divergence estimation and more stable training objectives in downstream machine-learning systems (e.g., generative modeling and distribution comparison). As with other generative and statistical modeling tools, the techniques studied here could be incorporated into applications that may generate or amplify harmful content or be used in settings with privacy, fairness, or security concerns. Our contribution does not introduce new data collection mechanisms, does not rely on sensitive attributes, and does not directly enable user identification; nevertheless, practitioners should follow established guidelines for responsible dataset use and deployment, and should evaluate downstream models for misuse and unintended consequences. The method is intended for divergence estimation and controlled critic-side regularization; deployment in generative systems should follow standard safeguards for dataset governance, misuse prevention, and privacy evaluation. Overall, we do not foresee specific broader societal harms uniquely attributable to the proposed estimator beyond those commonly associated with general-purpose generative modeling and statistical estimation methods.

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

# Appendix: Detailed Proofs and Additional Experiments

## A. Extended Related Work

Variational estimation with neural discriminators has become a cornerstone for approximating $\phi$-divergences and mutual information in modern machine learning. The $\phi$-GAN framework (Nowozin et al., 2016) formulates $\phi$-divergence estimation as a dual risk minimization problem over parameterized critics, enabling adversarial training across diverse divergence families. Similarly, the Mutual Information Neural Estimator (MINE) (Belghazi et al., 2018) employs a discriminator-based approach for scalable mutual information estimation via minibatch stochastic gradient descent. While these methods offer computational efficiency and flexibility, unconstrained neural critics frequently exhibit instability and high-variance estimates, even on simple distributions (Birrell et al., 2022; Sreekumar & Goldfeld, 2022). Recent work has demonstrated that these pathologies stem from insufficient control over discriminator complexity and smoothness (Ghimire et al., 2021), motivating the need for explicit capacity regularization in the critic's function class.

A complementary line of research constrains the critic to lie within a RKHS and explicitly penalizes the RKHS norm to achieve learning-theoretic control and finite-sample guarantees. Foundational work by (Nguyen et al., 2010; Sriperumbudur et al., 2012) established convex risk minimization frameworks with RKHS regularization for variational divergence functionals. More recently, (Ghimire et al., 2021) demonstrated that controlling RKHS smoothness stabilizes discriminator-based KL divergence estimation by bounding error probabilities in terms of the RKHS-ball radius and kernel complexity. Further theoretical advances (Zellinger et al., 2023) provide adaptive learning rates and finite-sample error bounds for density ratio estimation in RKHS. However, classical kernel methods rely on Gram matrix computations and the representer theorem, leading to $\mathcal{O}(n^2)$ memory complexity and $\mathcal{O}(n^3)$ time complexity for matrix operations (Bach & Jordan, 2005; Cesa-Bianchi et al., 2015), which severely limits scalability to large datasets.

RFF (Rahimi & Recht, 2007) provide a principled approach to approximating shift-invariant kernels by sampling frequencies from the kernel's spectral measure, as guaranteed by Bochner's theorem. The RFF construction maps inputs to finite-dimensional trigonometric feature vectors whose inner products yield unbiased kernel approximations. Theoretical analysis (Avron et al., 2017) shows that RFF approximation error decays exponentially in the number of features, with uniform convergence over compact sets. Recent extensions (He et al., 2024; Tóth & Oberhauser, 2025) generalize RFF to asymmetric kernels and signature-based features. By replacing Gram matrix operations with explicit low-dimensional feature maps, RFF reduces computational complexity to $\mathcal{O}(nm)$ for $n$ samples and $m$ features, enabling minibatch-friendly optimization (Zhang et al., 2019b). Empirically, RFF-based ridge regression achieves performance comparable to exact kernel methods while being substantially faster and more memory-efficient (Rahimi & Recht, 2007; Avron et al., 2017), making RFF an ideal vehicle for scaling RKHS-based divergence estimators to large-scale settings.

Within generative modeling, Amari's $\alpha$-divergence has emerged as a flexible family for stabilizing GAN training by tuning the parameter $\alpha$ to balance mode coverage and sample quality trade-offs. The Alpha-GAN framework (Cai et al., 2020) instantiates $\alpha$-divergence as the generator's objective function, analyzing hyperparameter choices for stability and perceptual quality across image generation benchmarks. Recent work (Veiner et al., 2024) unifies various GAN objectives under a parameterized generator loss function framework, demonstrating that different GAN variants correspond to specific choices within the $\phi$-divergence family. While these approaches highlight the benefits of divergence family flexibility, they typically optimize an *unconstrained* neural critic, leaving variance and overfitting concerns unresolved—precisely the regime where RKHS regularization combined with efficient kernel approximations can provide theoretical guarantees and empirical stability.

**Relation to WGAN and MMD-GAN.** WGAN and MMD-GAN are based on integral probability metrics of the form $\sup_f \mathbb{E}_P f - \mathbb{E}_Q f$ (Arjovsky et al., 2017; Bińkowski et al., 2018). Our estimator instead targets the Amari $\alpha$-divergence through the nonlinear power-type envelope $\mathbb{E}_P |f|^a - \mathbb{E}_Q |f|^b$. The shared functional-analytic perspective is useful: MMD-GAN also controls a kernel critic class, and WGAN studies critics in a constrained function space, including analyses of the optimal discriminator (Asokan & Seelamantula, 2023). The difference is the output and theory: we provide non-asymptotic estimation bounds, a feature-discretization/statistical/optimization decomposition, and local PL geometry for the RFF estimator, rather than a pure IPM training objective or a PDE-style characterization of an adversarial discriminator.

## B. Supplementary Background and Technical Derivations

### B.1. Mercer theory and the integral-operator viewpoint

Let $\mu$ be a reference measure on a compact $\mathcal{X}$ and let $k$ be continuous and positive definite. The associated integral operator $\mathcal{T}_k : L^2(\mu) \to L^2(\mu)$ is

$$(\mathcal{T}_k g)(x) = \int_{\mathcal{X}} k(x,z)\, g(z)\, d\mu(z).$$

It is compact, self-adjoint, and positive. Mercer's theorem implies that there exist eigenpairs $\{(\lambda_j, e_j)\}_{j \geq 1}$ with $\lambda_1 \geq \lambda_2 \geq \cdots \geq 0$ and $\{e_j\}$ orthonormal in $L^2(\mu)$ such that

$$k(x,z) = \sum_{j=1}^{\infty} \lambda_j e_j(x) e_j(z), \tag{31}$$

with uniform convergence on $\mathcal{X} \times \mathcal{X}$. This expansion yields a convenient spectral description of the RKHS norm: if $f = \sum_{j \geq 1} a_j e_j$, then

$$f \in \mathcal{H}_k \iff \sum_{j \geq 1} \frac{a_j^2}{\lambda_j} < \infty, \qquad \|f\|_k^2 = \sum_{j \geq 1} \frac{a_j^2}{\lambda_j}.$$

A common regularity assumption is a spectral source condition of the form $f^\star \in \mathrm{Range}(\mathcal{T}_k^r)$ for some $r > 0$, i.e., $f^\star = \mathcal{T}_k^r h$ with $h \in L^2(\mu)$, which implies the coefficients of $f^\star$ align with the eigenvalue decay $\{\lambda_j\}$. This operator viewpoint is also the basis for connecting random-feature approximations to truncated spectral expansions; see, e.g., (Bach, 2017).

### B.2. Variational envelope and optimal scaling

Fix $0 < a < b$ and define $A(f) := \mathbb{E}_P[|f|^a]$ and $B(f) := \mathbb{E}_Q[|f|^b]$. For any $f \not\equiv 0$, consider the scalarized objective

$$J(tf) = t^a A(f) - t^b B(f), \qquad t > 0.$$

Differentiating and solving $\frac{d}{dt} J(tf) = 0$ gives the unique maximizer

$$t^\star(f) = \left( \frac{a\, A(f)}{b\, B(f)} \right)^{\frac{1}{b-a}}. \tag{32}$$

Substituting back yields the closed-form envelope

$$\sup_{t > 0} J(tf) = C_{a,b}\, A(f)^{\frac{b}{b-a}}\, B(f)^{-\frac{a}{b-a}}, \qquad C_{a,b} := \frac{b-a}{b} \left( \frac{a}{b} \right)^{\frac{a}{b-a}}. \tag{33}$$

With $\alpha = b/(b-a)$, maximizing this envelope over nonzero $f$ yields (4) in the main text, which provides the variational bridge between the critic maximum and the $\alpha$-affinity $I_\alpha(P,Q)$ and thus $D_\alpha^{(A)}(P\|Q)$.

### B.3. Smoothing and the gradient used in optimization

To ensure differentiability at the origin and to stabilize optimization when $a$ or $b$ is below 1, we replace $|u|^p$ by a smooth surrogate

$$\phi_{p,\varepsilon}(u) := (u^2 + \varepsilon^2)^{p/2}, \qquad \varepsilon > 0.$$

Its derivative is

$$\frac{d}{du} \phi_{p,\varepsilon}(u) = p\, (u^2 + \varepsilon^2)^{\frac{p}{2}-1}\, u,$$

which directly yields the gradient formula (9) for the smoothed empirical objective (8). The surrogate also introduces a controlled approximation error: since $(u^2 + \varepsilon^2)^{1/2} \geq |u|$, one has $|u|^p \leq (u^2 + \varepsilon^2)^{p/2}$, and for fixed $p > 0$ the discrepancy vanishes uniformly as $\varepsilon \to 0$ on bounded sets. In our analysis, $\varepsilon$ is treated as a small numerical smoothing parameter; the statistical and optimization guarantees are derived for the smoothed objective and then related back to the original power objective through this approximation.

## B.4. Alternative parameterization of Amari $\alpha$-divergence

A common information-geometric parameterization uses $\tilde{\alpha} \in (-1, 1)$ and defines

$$D_A^{(\tilde{\alpha})}(P\|Q) = \frac{4}{1-\tilde{\alpha}^2}\left(1 - \int_{\mathcal{X}} p(x)^{\frac{1+\tilde{\alpha}}{2}} q(x)^{\frac{1-\tilde{\alpha}}{2}} \, dx\right). \tag{34}$$

The two parameterizations are equivalent under the reparameterization $\alpha = \frac{1+\tilde{\alpha}}{2}$.

**Lemma B.1** (Equivalence of parameterizations). *Let $\tilde{\alpha} \in (-1, 1)$ and set $\alpha = \frac{1+\tilde{\alpha}}{2} \in (0,1)$. Then $D_A^{(\tilde{\alpha})}(P\|Q) = D_A^{(\alpha)}(P\|Q)$ where $D_A^{(\alpha)}$ is defined in (1).*

*Proof.* With $\alpha = \frac{1+\tilde{\alpha}}{2}$ we have $1 - \alpha = \frac{1-\tilde{\alpha}}{2}$ and $\alpha(1-\alpha) = \frac{(1+\tilde{\alpha})(1-\tilde{\alpha})}{4} = \frac{1-\tilde{\alpha}^2}{4}$. Substituting these identities into (1) gives

$$D_A^{(\alpha)}(P\|Q) = \frac{4}{1-\tilde{\alpha}^2}\left(1 - \int p(x)^{\frac{1+\tilde{\alpha}}{2}} q(x)^{\frac{1-\tilde{\alpha}}{2}} \, dx\right),$$

which matches (34). $\qquad\square$

## B.5. Information-geometric structure of the $\alpha$-family

We briefly record the geometric facts behind the statement that the Amari $\alpha$-family sits at the intersection of $f$-divergences and Bregman divergences; see (Amari, 2009) for a comprehensive treatment.

Let $\mathcal{P}$ be a statistical manifold (a smooth family of strictly positive densities), and let $D(P\|Q)$ be a smooth divergence on $\mathcal{P} \times \mathcal{P}$. The divergence induces a Riemannian metric and dual affine connections by taking derivatives along the diagonal:

$$g_{ij}(\theta) := \left.\frac{\partial^2}{\partial\theta^i \partial\theta'^j} D\big(P_\theta \| P_{\theta'}\big)\right|_{\theta'=\theta}, \tag{35}$$

$$\Gamma_{ij,k}(\theta) := -\left.\frac{\partial^3}{\partial\theta^i \partial\theta^j \partial\theta'^k} D\big(P_\theta \| P_{\theta'}\big)\right|_{\theta'=\theta}, \qquad \Gamma_{ij,k}^*(\theta) := -\left.\frac{\partial^3}{\partial\theta'^i \partial\theta'^j \partial\theta^k} D\big(P_\theta \| P_{\theta'}\big)\right|_{\theta'=\theta}. \tag{36}$$

For the Amari $\alpha$-divergence, the induced metric $g$ coincides with the Fisher information metric, and the induced connections form the $\alpha$-connection family, with $(\nabla^{(\alpha)})^* = \nabla^{(1-\alpha)}$ under the normalization (1). This yields the duality relation $D_A^{(\alpha)}(P\|Q) = D_A^{(1-\alpha)}(Q\|P)$ already stated in the main text, and it is the precise sense in which the $\alpha$-family provides a canonical interpolation between the forward and reverse KL limits. When one switches to the $\tilde{\alpha}$ convention (34), the same objects are commonly expressed with $\tilde{\alpha} \in (-1, 1)$ and $\alpha = \frac{1+\tilde{\alpha}}{2}$. These constructions are background only; our analysis uses (1) and its variational consequences.

# C. Auxiliary Results for Section 4

## C.1. Algebraic properties of $\phi_{p,\varepsilon}$

Recall $\phi_{p,\varepsilon}(t) = (t^2 + \varepsilon^2)^{p/2}$ for $p > 0$ and $\varepsilon > 0$. The proxy gap (14) follows from the elementary bounds

$$0 \leq |t|^p - \phi_{p,\varepsilon}(t) \leq \varepsilon^p, \qquad \forall t \in \mathbb{R},$$

which hold because $(t^2 + \varepsilon^2)^{1/2} \in [|t|, |t| + \varepsilon]$ and the map $u \mapsto u^p$ is nondecreasing on $\mathbb{R}_+$.

## C.2. Uniform control of empirical mismatch

Let $V := V_D \cap \mathcal{B}_R$ and define the uniform proxy generalization gap

$$\mathcal{E}_{\mathrm{gen}} := \sup_{g \in V} \left|J_\varepsilon(g) - \widehat{J}_{n,\varepsilon}(g)\right|.$$

Let $g_{\varepsilon,D}^\star \in \arg\max_{v \in V} J_\varepsilon(v)$ and $\widehat{g}_{D,R}^\star \in \arg\max_{v \in V} \widehat{J}_{n,\varepsilon}(v)$.

**Lemma C.1** (Uniform control of empirical mismatch). *With $\mathcal{E}_{\mathrm{mis}} := \widehat{J}_{n,\varepsilon}(g_{D,R}^{\dagger}) - \widehat{J}_{n,\varepsilon}(\widehat{g}_{D,R}^{\star}) \leq 0$, it holds that*

$$\mathcal{E}_{\mathrm{mis}} \leq 2\,\mathcal{E}_{\mathrm{gen}}.$$

*Consequently,*

$$\mathcal{E}_{\mathrm{stat}} + \mathcal{E}_{\mathrm{mis}} \leq 3\,\mathcal{E}_{\mathrm{gen}}, \qquad \mathcal{E}_{\mathrm{stat}} := J_{\varepsilon}(g_{D,R}^{\dagger}) - \widehat{J}_{n,\varepsilon}(g_{D,R}^{\dagger}).$$

*Proof.* Since $\widehat{g}_{D,R}^{\star}$ maximizes $\widehat{J}_{n,\varepsilon}$ over $V$,

$$\mathcal{E}_{\mathrm{mis}} \leq \widehat{J}_{n,\varepsilon}(g_{D,R}^{\dagger}) - \widehat{J}_{n,\varepsilon}(g_{\varepsilon,D}^{\star}).$$

Add and subtract $J_{\varepsilon}$ at both points:

$$\begin{aligned}
\widehat{J}_{n,\varepsilon}(g_{D,R}^{\dagger}) - \widehat{J}_{n,\varepsilon}(g_{\varepsilon,D}^{\star}) &= \left[ J_{\varepsilon}(g_{D,R}^{\dagger}) - J_{\varepsilon}(g_{\varepsilon,D}^{\star}) \right] \\
&\quad + \left[ \widehat{J}_{n,\varepsilon}(g_{D,R}^{\dagger}) - J_{\varepsilon}(g_{D,R}^{\dagger}) \right] + \left[ J_{\varepsilon}(g_{\varepsilon,D}^{\star}) - \widehat{J}_{n,\varepsilon}(g_{\varepsilon,D}^{\star}) \right] \\
&\leq 0 + 2\,\mathcal{E}_{\mathrm{gen}},
\end{aligned}$$

because $g_{\varepsilon,D}^{\star}$ maximizes $J_{\varepsilon}$ over $V$. Finally, $\mathcal{E}_{\mathrm{stat}} \leq \mathcal{E}_{\mathrm{gen}}$ by definition, so $\mathcal{E}_{\mathrm{stat}} + \mathcal{E}_{\mathrm{mis}} \leq \mathcal{E}_{\mathrm{gen}} + 2\mathcal{E}_{\mathrm{gen}} = 3\mathcal{E}_{\mathrm{gen}}$. $\qquad\square$

# D. Proofs for Function Approximation in RKHS

This appendix provides the detailed proofs for the approximation guarantees within the RKHS used in Section 4. We prove the $L^2$ truncation rate (Theorem 4.3) and then upgrade it to an $L^\infty$ rate via Sobolev interpolation (Lemma D.2).

## D.1. Proof of Theorem 4.3

*Remark* D.1 (Gaussian kernel example for Assumption 4.1). For the Gaussian kernel with variance $\sigma^2$, $\hat{k}(\omega) = (2\pi\sigma^2)^{d/2} \exp(-\sigma^2\|\omega\|^2/2)$. Hence on $\{|\omega| \leq W\}$ one may take $M_k(W) = (2\pi\sigma^2)^{d/2}$ and $m_k(W) = (2\pi\sigma^2)^{d/2} \exp(-\sigma^2 W^2/2)$.

*Proof.* The construction proceeds by truncating $\widehat{f^\star}$ and then choosing $W$ to saturate the RKHS budget. For any $W > 0$, define $g_W$ by

$$\widehat{g_W}(\omega) := \widehat{f^\star}(\omega)\,\mathbb{I}(|\omega| \leq W).$$

By Plancherel and Assumption 4.2,

$$\|f^\star - g_W\|_{L^2(\mathbb{R}^d)}^2 = \int_{|\omega|>W} \hat{k}(\omega)^{2r} |\widehat{\xi}(\omega)|^2 \, d\omega.$$

Since $\hat{k}$ is radially nonincreasing, for $|\omega| > W$ we have $\hat{k}(\omega) \leq \hat{k}(W) \leq m_k(W)$, hence

$$\begin{aligned}
\|f^\star - g_W\|_{L^2(\mathbb{R}^d)}^2 &\leq m_k(W)^{2r} \int_{\mathbb{R}^d} |\widehat{\xi}(\omega)|^2 \, d\omega \\
&= m_k(W)^{2r} \|\xi\|_{L^2(\mathbb{R}^d)}^2 \leq m_k(W)^{2r} M_\xi^2.
\end{aligned} \tag{37}$$

Taking square roots yields

$$\|f^\star - g_W\|_{L^2(\mathbb{R}^d)} \leq M_\xi\, m_k(W)^r. \tag{38}$$

Next, by the Fourier characterization of $\mathcal{H}_k$ and $\widehat{f^\star} = \hat{k}^r \widehat{\xi}$,

$$\|g_W\|_{\mathcal{H}_k}^2 = \int_{|\omega|\leq W} \frac{|\widehat{f^\star}(\omega)|^2}{\hat{k}(\omega)} \, d\omega = \int_{|\omega|\leq W} \hat{k}(\omega)^{2r-1} |\widehat{\xi}(\omega)|^2 \, d\omega.$$

Since $2r - 1 < 0$ and $\hat{k}(\omega) \geq m_k(W)$ for $|\omega| \leq W$ (Assumption 4.1),

$$
\begin{aligned}
\|g_W\|_{\mathcal{H}_k}^2 &\leq m_k(W)^{2r-1} \int_{\mathbb{R}^d} |\widehat{\xi}(\omega)|^2 \, d\omega \\
&= m_k(W)^{2r-1} \|\xi\|_{L^2(\mathbb{R}^d)}^2 \ \leq \ m_k(W)^{2r-1} M_\xi^2.
\end{aligned}
\tag{39}
$$

Fix $R > 0$ and choose $W = W(R)$ so that the budget holds, e.g., $m_k(W(R))^{2r-1} M_\xi^2 \leq R^2$. Let $\alpha := \frac{1}{2} - r > 0$. This condition is equivalent to

$$
m_k(W(R)) \geq \left(\frac{R}{M_\xi}\right)^{-\frac{1}{\alpha}}.
$$

Plugging into (38) gives

$$
\begin{aligned}
\|f^\star - g_{W(R)}\|_{L^2(\mathbb{R}^d)} &\leq M_\xi \, m_k(W(R))^r \\
&\leq M_\xi \left(\frac{R}{M_\xi}\right)^{-\frac{r}{\alpha}} \ = \ C_1 \, R^{-\frac{r}{\alpha}} \ = \ C_1 \, R^{-\frac{2r}{1-2r}},
\end{aligned}
$$

where $C_1$ depends on $(r, M_\xi)$ and the spectrum through $m_k(\cdot)$. Setting $g_R := g_{W(R)}$ completes the proof. $\qquad\square$

### D.2. Sobolev interpolation: upgrading $L^2$ to $L^\infty$

**Lemma D.2** (Gagliardo–Nirenberg/Sobolev interpolation). *Let $\mathcal{M} \subset \mathbb{R}^d$ be a compact domain with Lipschitz boundary and let $s > d/2$. There exists a constant $C_{\mathrm{GN}}$ (depending on $\mathcal{M}, d, s$) such that for all $u \in H^s(\mathcal{M})$,*

$$
\|u\|_{L^\infty(\mathcal{M})} \ \leq \ C_{\mathrm{GN}} \|u\|_{L^2(\mathcal{M})}^{1-\theta} \|u\|_{H^s(\mathcal{M})}^{\theta}, \qquad \theta := \frac{d}{2s} \in (0,1).
\tag{40}
$$

*Proof.* This is a standard consequence of the Sobolev embedding $H^s(\mathcal{M}) \hookrightarrow L^\infty(\mathcal{M})$ for $s > d/2$ together with interpolation between $L^2$ and $H^s$ (equivalently, a Gagliardo–Nirenberg inequality). $\qquad\square$

*Proof of the $L^\infty$ rate used in* (18). Let $u_R := f^\star - g_R$ and assume $\|u_R\|_{H^s(\mathcal{M})} \leq K_s$ for some $s > d/2$. From Theorem 4.3 we have $\|u_R\|_{L^2(\mathcal{M})} \leq C_1 R^{-\beta}$ with $\beta = \frac{2r}{1-2r}$. Applying Lemma D.2 yields

$$
\begin{aligned}
\|u_R\|_{L^\infty(\mathcal{M})} &\leq C_{\mathrm{GN}} \, (C_1 R^{-\beta})^{1-\theta} \, K_s^\theta \\
&= \left(C_{\mathrm{GN}} C_1^{1-\theta} K_s^\theta\right) R^{-\beta(1-\theta)} \ = \ C_\infty \, R^{-\gamma},
\end{aligned}
$$

where $\gamma = \beta(1-\theta) = \frac{2r}{1-2r}(1 - \frac{d}{2s})$. $\qquad\square$

## E. Proofs for Feature Approximation with RFF

This appendix provides the derivation of the RFF approximation bound tailored to bandlimited RKHS functions used in Section 4.

**Lemma E.1** (RFF approximation for bandlimited RKHS functions). *Let $\mathcal{M} \subset \mathbb{R}^d$ be compact and let $k$ be a bounded shift-invariant kernel with spectral measure $\Lambda$. Fix $R > 0$ and a window $W > 0$, and let $\Lambda_W$ be the normalized truncation of $\Lambda$ to the ball $B_W := \{\omega : \|\omega\| \leq W\}$:*

$$
\Lambda_W(A) := \frac{\Lambda(A \cap B_W)}{\Lambda(B_W)} \qquad (A \text{ Borel}).
$$

*Assume $h \in \mathcal{H}_k$ satisfies $\|h\|_{\mathcal{H}_k} \leq R$ and $\widehat{h}(\omega) = 0$ for $\|\omega\| > W$. Then there exists a constant $C_{\mathrm{RFF}} > 0$ depending only on $k$ and $\mathrm{diam}(\mathcal{M})$ such that, for any $\delta \in (0,1)$, with probability at least $1 - \delta$ over $\omega_{1:D} \sim \Lambda_W$, there exists $v_{D,W} \in V_D$ with*

$$
\|h - v_{D,W}\|_{L^\infty(\mathcal{M})} \ \leq \ C_{\mathrm{RFF}} \, R \left( W \sqrt{\frac{d}{D}} + \sqrt{\frac{\log(1/\delta)}{D}} \right).
\tag{41}
$$

### E.1. Proof of Lemma E.1

*Proof.* We control the uniform Monte–Carlo approximation error by a standard empirical-process argument for RFF; see (Sriperumbudur & Szabó, 2015) for closely related uniform bounds.

Because $h$ is bandlimited to $B_W := \{\omega : \|\omega\| \leq W\}$ and $k$ is shift-invariant, there exists $a \in L^2(\Lambda)$ supported on $B_W$ such that

$$h(x) = \int_{B_W} a(\omega)\, e^{i\omega^\top x}\, d\Lambda(\omega), \qquad \|h\|_{\mathcal{H}_k}^2 = \int_{B_W} \frac{|a(\omega)|^2}{d\Lambda(\omega)}\, d\Lambda(\omega) \leq R^2.$$

Let $\Lambda_W$ denote the normalized truncation of $\Lambda$ on $B_W$, namely $\Lambda_W(A) = \Lambda(A \cap B_W)/\Lambda(B_W)$. Then

$$h(x) = \Lambda(B_W) \int a(\omega) e^{i\omega^\top x}\, d\Lambda_W(\omega), \qquad \|a\|_{L^2(\Lambda_W)} \leq \frac{R}{\sqrt{\Lambda(B_W)}}.$$

Draw $\omega_1, \ldots, \omega_D \overset{\text{i.i.d.}}{\sim} \Lambda_W$ and define the Monte–Carlo approximant

$$\widetilde{v}_{D,W}(x) := \Lambda(B_W) \cdot \frac{1}{D} \sum_{j=1}^{D} a(\omega_j) e^{i\omega_j^\top x}.$$

Taking real and imaginary parts shows that $\widetilde{v}_{D,W}$ lies in the span of the cosine–sine random features, so there exists $v_{D,W} \in V_D$ such that $\|h - v_{D,W}\|_{L^\infty(\mathcal{M})} \leq \|h - \widetilde{v}_{D,W}\|_{L^\infty(\mathcal{M})}$. It therefore suffices to bound $\|h - \widetilde{v}_{D,W}\|_{L^\infty(\mathcal{M})}$.

For each $x \in \mathcal{M}$ define $g_x(\omega) := \Lambda(B_W) a(\omega) e^{i\omega^\top x}$ on $B_W$, and let $\Lambda_{W,D} := \frac{1}{D} \sum_{j=1}^{D} \delta_{\omega_j}$. Then $h(x) = \int g_x\, d\Lambda_W$ and $\widetilde{v}_{D,W}(x) = \int g_x\, d\Lambda_{W,D}$, hence

$$\|h - \widetilde{v}_{D,W}\|_{L^\infty(\mathcal{M})} = \sup_{x \in \mathcal{M}} \left| \big(\Lambda_W - \Lambda_{W,D}\big) g_x \right| =: F(\omega_{1:D}).$$

A bounded-differences concentration argument applies to $F$. Replacing one frequency $\omega_i$ by an independent copy $\omega_i'$ changes $\Lambda_{W,D}$ by at most $1/D$, which implies

$$|F(\omega_{1:D}) - F(\omega_1, \ldots, \omega_i', \ldots, \omega_D)| \leq \frac{1}{D} \sup_{x \in \mathcal{M}} |g_x(\omega_i) - g_x(\omega_i')| \leq \frac{2\Lambda(B_W)}{D} \max\{|a(\omega_i)|, |a(\omega_i')|\}.$$

Using the square-integrable envelope control induced by $\|a\|_{L^2(\Lambda_W)} < \infty$ and the same truncation/variance handling as in the RFF uniform analysis of (Sriperumbudur & Szabó, 2015), one obtains that for any $\delta \in (0,1)$, with probability at least $1 - \delta$,

$$F(\omega_{1:D}) \leq \mathbb{E} F(\omega_{1:D}) + C_1\, \Lambda(B_W)\, \|a\|_{L^2(\Lambda_W)} \sqrt{\frac{\log(1/\delta)}{D}}, \tag{42}$$

for an absolute constant $C_1 > 0$.

To bound $\mathbb{E} F$, symmetrization yields

$$\mathbb{E} F(\omega_{1:D}) \leq 2\, \mathbb{E}\, \mathbb{E}_\varepsilon \sup_{x \in \mathcal{M}} \left| \frac{1}{D} \sum_{j=1}^{D} \varepsilon_j g_x(\omega_j) \right|,$$

where $\varepsilon_1, \ldots, \varepsilon_D$ are i.i.d. Rademacher variables. Conditioned on $\omega_{1:D}$, the Rademacher process is sub-Gaussian and admits Dudley's entropy integral bound in terms of covering numbers of the function class $\mathcal{G} := \{g_x : x \in \mathcal{M}\}$ equipped with the metric $L^2(\Lambda_{W,D})$, as in (Sriperumbudur & Szabó, 2015). It therefore remains to control $N(\mathcal{G}, L^2(\Lambda_{W,D}), r)$.

For $x, x' \in \mathcal{M}$ and $\omega \in B_W$, $|e^{i\omega^\top x} - e^{i\omega^\top x'}| \leq \|\omega\|\|x - x'\| \leq W\|x - x'\|$, which implies

$$\|g_x - g_{x'}\|_{L^2(\Lambda_{W,D})}^2 = \frac{\Lambda(B_W)^2}{D} \sum_{j=1}^{D} |a(\omega_j)|^2\, |e^{i\omega_j^\top x} - e^{i\omega_j^\top x'}|^2 \leq \Lambda(B_W)^2 W^2 \Big( \frac{1}{D} \sum_{j=1}^{D} |a(\omega_j)|^2 \Big) \|x - x'\|^2.$$

Consequently, any Euclidean $\varepsilon$-net of $\mathcal{M}$ induces an $r$-net of $\mathcal{G}$ in $L^2(\Lambda_{W,D})$ with $r = \Lambda(B_W)W\big(\frac{1}{D}\sum_{j=1}^{D}|a(\omega_j)|^2\big)^{1/2}\varepsilon$. Since $\mathcal{M} \subset \mathbb{R}^d$ is compact, the volumetric bound gives

$$\log N(\mathcal{G}, L^2(\Lambda_{W,D}), r) \leq d\log\Big(1 + \frac{C_2\,\Lambda(B_W)\,W}{r}\Big(\frac{1}{D}\sum_{j=1}^{D}|a(\omega_j)|^2\Big)^{1/2}\Big),$$

where $C_2 > 0$ depends only on $\mathrm{diam}(\mathcal{M})$. Evaluating Dudley's integral with this entropy bound yields a constant $C_3 > 0$, depending only on $k$ and $\mathrm{diam}(\mathcal{M})$, such that

$$\mathbb{E}F(\omega_{1:D}) \leq C_3\,\Lambda(B_W)\,W\,\|a\|_{L^2(\Lambda_W)}\sqrt{\frac{d}{D}}.$$

Combining this with (42) gives that, with probability at least $1 - \delta$,

$$\|h - \widetilde{v}_{D,W}\|_{L^\infty(\mathcal{M})} \leq C_4\,\Lambda(B_W)\,\|a\|_{L^2(\Lambda_W)}\left(W\sqrt{\frac{d}{D}} + \sqrt{\frac{\log(1/\delta)}{D}}\right),$$

for a constant $C_4 > 0$ depending only on $k$ and $\mathrm{diam}(\mathcal{M})$. Finally, using $\|a\|_{L^2(\Lambda_W)} \leq R/\sqrt{\Lambda(B_W)}$ and absorbing $\Lambda(B_W)$ into the constant yields

$$\|h - \widetilde{v}_{D,W}\|_{L^\infty(\mathcal{M})} \leq C_{\mathrm{RFF}}\,R\left(W\sqrt{\frac{d}{D}} + \sqrt{\frac{\log(1/\delta)}{D}}\right).$$

Since $\|h - v_{D,W}\|_\infty \leq \|h - \widetilde{v}_{D,W}\|_\infty$ for some $v_{D,W} \in V_D$, the claim follows. $\square$

# F. Proofs for Statistical Error Analysis

This appendix provides the detailed proofs for the statistical error bounds. The key technical input is an entropy bound for bandlimited RKHS balls (Lemma F.1), which is then combined with chaining to prove Theorem 4.4.

**Lemma F.1** (Entropy of bandlimited RKHS balls). *Let $k$ be a bounded, shift-invariant kernel on $\mathbb{R}^d$ with spectral density $\hat{k}$ satisfying Assumption 4.1. Fix a spectral window $W \geq W_0$ and define the bandlimited RKHS ball*

$$\mathcal{B}_{R,W}^{\mathrm{BL}} := \big\{g \in \mathcal{H}_{k,W} : \|g\|_{\mathcal{H}_k} \leq R\big\}, \qquad \mathcal{H}_{k,W} := \big\{g \in \mathcal{H}_k : \widehat{g}(\omega) = 0\,\text{for}\,\|\omega\| > W\big\}.$$

*Then there exist $h > 0$ and $C_h(W) > 0$, depending only on $(k, W, \mathcal{M})$, such that for all $R > 0$ and all $0 < \eta \leq \sqrt{S_k}\,R$,*

$$\ln\mathcal{N}\big(\mathcal{B}_{R,W}^{\mathrm{BL}}, \eta, \|\cdot\|_\infty\big) \leq \big(C_h(W)\,R\,\sqrt{S_k}/\eta\big)^{2d/h},$$

*where $S_k := \sup_{x \in \mathcal{M}} k(x,x)$ and $\mathcal{N}(\cdot, \eta, \|\cdot\|_\infty)$ is the $\eta$-covering number in supremum norm. Moreover, any $V_D \subset \mathcal{H}_{k,W}$ with $\|g\|_{\mathcal{H}_k} \leq R$ for all $g \in V_D$ satisfies*

$$\ln\mathcal{N}\big(V_D, \eta, \|\cdot\|_\infty\big) \leq \ln\mathcal{N}\big(\mathcal{B}_{R,W}^{\mathrm{BL}}, \eta, \|\cdot\|_\infty\big)$$

*for the same constants.*

## F.1. Proof of Lemma F.1

*Proof.* We First state that bandlimited RKHS functions have bounded Sobolev norms and then get sobolev embedding which is connected with smoothness parameter. By shift-invariance and Bochner's theorem, any $g \in \mathcal{H}_k$ admits a Fourier representation

$$\|g\|_{\mathcal{H}_k}^2 = \int_{\mathbb{R}^d} \frac{|\widehat{g}(\omega)|^2}{\hat{k}(\omega)}\,d\omega,$$

with $\widehat{g}$ supported in $\{\|\omega\| \leq W\}$ whenever $g \in \mathcal{H}_{k,W}$. Under Assumption 4.1, for all $\|\omega\| \leq W$,

$$0 < m_k(W) \leq \hat{k}(\omega) \leq M_k(W) < \infty.$$

Hence, for any $g \in \mathcal{H}_{k,W}$,

$$\int_{\mathbb{R}^d} |\widehat{g}(\omega)|^2 \, d\omega \;=\; \int_{\|\omega\| \leq W} |\widehat{g}(\omega)|^2 \, d\omega \;\leq\; M_k(W) \, \|g\|_{\mathcal{H}_k}^2. \tag{43}$$

Fix an integer $m \geq 1$ (to be chosen later). For a multi-index $\alpha \in \mathbb{N}^d$ with $|\alpha| \leq m$, the Fourier transform of the weak derivative $\partial^\alpha g$ is

$$\widehat{\partial^\alpha g}(\omega) = (i\omega)^\alpha \widehat{g}(\omega),$$

hence, by Plancherel,

$$\begin{aligned}
\|\partial^\alpha g\|_{L^2(\mathbb{R}^d)}^2 &= \int_{\mathbb{R}^d} |\widehat{\partial^\alpha g}(\omega)|^2 \, d\omega \\
&= \int_{\|\omega\| \leq W} |\omega|^{2|\alpha|} |\widehat{g}(\omega)|^2 \, d\omega \\
&\leq W^{2|\alpha|} \int_{\|\omega\| \leq W} |\widehat{g}(\omega)|^2 \, d\omega \\
&\leq W^{2|\alpha|} M_k(W) \, \|g\|_{\mathcal{H}_k}^2.
\end{aligned} \tag{44}$$

using (43) in the last step.

Restricting to the compact domain $\mathcal{M} \subset \mathbb{R}^d$ only decreases the $L^2$-norm. Thus, for all $|\alpha| \leq m$,

$$\|\partial^\alpha g\|_{L^2(\mathcal{M})} \;\leq\; \|\partial^\alpha g\|_{L^2(\mathbb{R}^d)} \;\leq\; W^{|\alpha|} \sqrt{M_k(W)} \, \|g\|_{\mathcal{H}_k}.$$

Define the Sobolev $H^m(\mathcal{M})$-norm by

$$\|g\|_{H^m(\mathcal{M})} := \left( \sum_{|\alpha| \leq m} \|\partial^\alpha g\|_{L^2(\mathcal{M})}^2 \right)^{1/2}.$$

Summing the bounds above over $|\alpha| \leq m$, we obtain

$$\begin{aligned}
\|g\|_{H^m(\mathcal{M})} &\leq \left( \sum_{|\alpha| \leq m} W^{2|\alpha|} M_k(W) \right)^{1/2} \|g\|_{\mathcal{H}_k} \\
&\leq C_1(W, m) \, \|g\|_{\mathcal{H}_k},
\end{aligned} \tag{45}$$

where

$$C_1(W, m) := \left( M_k(W) \sum_{|\alpha| \leq m} W^{2|\alpha|} \right)^{1/2}$$

depends only on $(W, m, k)$. In particular, every $g \in \mathcal{H}_{k,W}$ with $\|g\|_{\mathcal{H}_k} \leq R$ satisfies

$$\|g\|_{H^m(\mathcal{M})} \;\leq\; C_1(W, m) \, R.$$

Thus

$$\mathcal{B}_{R,W}^{\mathrm{BL}} \;\subset\; \{ g : \|g\|_{H^m(\mathcal{M})} \leq C_1(W, m) \, R \}. \tag{46}$$

By the reproducing property in the RKHS,

$$
\begin{aligned}
\|\partial^\alpha g\|&_{L^2(\mathbb{R}^d)}^2 \\
&= \int_{\mathbb{R}^d} |\widehat{\partial^\alpha g}(\omega)|^2 \, d\omega \\
&= \int_{\|\omega\| \le W} |\omega|^{2|\alpha|} |\widehat{g}(\omega)|^2 \, d\omega \\
&\le W^{2|\alpha|} \int_{\|\omega\| \le W} |\widehat{g}(\omega)|^2 \, d\omega \\
&\le W^{2|\alpha|} M_k(W) \|g\|_{\mathcal{H}_k}^2.
\end{aligned}
\tag{47}
$$

where $S_k := \sup_{x \in \mathcal{M}} k(x, x)$. Therefore, for all $g \in \mathcal{B}_{R,W}^{\mathrm{BL}}$,

$$
\|g\|_\infty \;\le\; \sqrt{S_k}\,\|g\|_{\mathcal{H}_k} \;\le\; \sqrt{S_k}\,R.
\tag{48}
$$

In particular, if $\eta > \sqrt{S_k}R$, then a single function (e.g. 0) covers the ball in sup-norm and $\mathcal{N} = 1$; hence the lemma is only nontrivial for $0 < \eta \le \sqrt{S_k}R$, which is exactly the range stated.

Next, recall the Sobolev embedding theorem on a bounded Lipschitz domain $\mathcal{M} \subset \mathbb{R}^d$: if $m > d/2$, then

$$
H^m(\mathcal{M}) \hookrightarrow C^0(\mathcal{M})
$$

continuously, i.e., there exists $C_{\mathrm{emb}}(\mathcal{M}, m) > 0$ such that

$$
\|g\|_\infty \;\le\; C_{\mathrm{emb}}(\mathcal{M}, m)\,\|g\|_{H^m(\mathcal{M})} \qquad \forall g \in H^m(\mathcal{M}).
\tag{49}
$$

Combining (45) and (49), we see that for all $g \in \mathcal{B}_{R,W}^{\mathrm{BL}}$,

$$
\begin{aligned}
\|g\|&_\infty \\
&\le C_{\mathrm{emb}}(\mathcal{M}, m) C_1(W, m) \|g\|_{\mathcal{H}_k} \\
&\le C_{\mathrm{emb}}(\mathcal{M}, m) C_1(W, m) R.
\end{aligned}
$$

This gives a second (redundant but harmless) envelope bound; more importantly, it shows that the inclusion (46) respects the sup-norm: the $H^m$-ball on the right is mapped continuously into $(C(\mathcal{M}), \|\cdot\|_\infty)$. Consider the Sobolev ball

$$
\begin{aligned}
\mathcal{B}_{R_S}^{(m)} &:= \{g \in H^m(\mathcal{M}) : \|g\|_{H^m(\mathcal{M})} \le R_S\} \\
&\text{with} \quad R_S := C_1(W, m)\,R.
\end{aligned}
$$

By (46),

$$
\mathcal{B}_{R,W}^{\mathrm{BL}} \subset \mathcal{B}_{R_S}^{(m)}.
$$

Since covering numbers decrease under set inclusion,

$$
\begin{aligned}
\mathcal{N}\big(&\mathcal{B}_{R,W}^{\mathrm{BL}}, \eta, \|\cdot\|_\infty\big) \\
&\le \mathcal{N}\big(\mathcal{B}_{R_S}^{(m)}, \eta, \|\cdot\|_\infty\big), \\
&\text{for } 0 < \eta \le \sqrt{S_k}R.
\end{aligned}
\tag{50}
$$

It remains to bound the right-hand side. Classical entropy results for Sobolev balls (see, e.g., Theorem 2.7.1 in van der Vaart and Wellner, *Weak Convergence and Empirical Processes*, 1996, adapted from Kolmogorov–Tikhomirov and Birman–Solomjak) state that for $m > d/2$ there exists a constant $C_0(\mathcal{M}, m) > 0$ such that

$$
\ln \mathcal{N}\big(\mathcal{B}_1^{(m)}, \eta, \|\cdot\|_\infty\big) \;\le\; C_0(\mathcal{M}, m)\,\eta^{-d/m}, \qquad 0 < \eta \le 1.
\tag{51}
$$

By homogeneity of the Sobolev norm and the sup-norm, rescaling from radius 1 to $R_S$ yields

$$\mathcal{N}\big(\mathcal{B}_{R_S}^{(m)}, \eta, \|\cdot\|_\infty\big) = \mathcal{N}\big(\mathcal{B}_1^{(m)}, \eta/R_S, \|\cdot\|_\infty\big),$$

hence, whenever $\eta \leq R_S$,

$$\ln \mathcal{N}\big(\mathcal{B}_{R_S}^{(m)}, \eta, \|\cdot\|_\infty\big) \leq C_0(\mathcal{M}, m)\, (R_S/\eta)^{d/m}. \tag{52}$$

We now set $h := 2m$, so that

$$\frac{2d}{h} = \frac{2d}{2m} = \frac{d}{m},$$

and note that $m > d/2$ implies $h > d$. Using $R_S = C_1(W, m)\, R$, we can rewrite (52) as

$$\ln \mathcal{N}\big(\mathcal{B}_{R_S}^{(m)}, \eta, \|\cdot\|_\infty\big) \leq \Big(C_h(W)\, R/\eta\Big)^{2d/h},$$

where we define

$$C_h(W) := \Big(C_0(\mathcal{M}, m)\Big)^{h/(2d)} \Big(C_1(W, m)\Big)^{d/m},$$

which depends only on $(k, W, \mathcal{M})$ through $M_k(W)$, $W$, $m$, and geometric properties of $\mathcal{M}$. Finally, recalling that $0 < \eta \leq \sqrt{S_k}R$, we absorb the factor $\sqrt{S_k}$ into the constant by writing

$$\frac{R}{\eta} = \frac{R\sqrt{S_k}}{\eta} \cdot \frac{1}{\sqrt{S_k}} \leq \frac{R\sqrt{S_k}}{\eta},$$

and enlarging $C_h(W)$ if necessary to obtain

$$\ln \mathcal{N}\big(\mathcal{B}_{R_S}^{(m)}, \eta, \|\cdot\|_\infty\big) \leq \Big(C_h(W)\, R\, \sqrt{S_k}/\eta\Big)^{2d/h}.$$

Combining this with (50) gives the claimed bound

$$\ln \mathcal{N}\big(\mathcal{B}_{R,W}^{\mathrm{BL}}, \eta, \|\cdot\|_\infty\big) \leq \Big(C_h(W)\, R\, \sqrt{S_k}/\eta\Big)^{2d/h}.$$

If $V_D \subset \mathcal{H}_{k,W}$ with $\|g\|_{\mathcal{H}_k} \leq R$ for all $g \in V_D$, then

$$V_D \subset \mathcal{B}_{R,W}^{\mathrm{BL}},$$

and by monotonicity of covering numbers under set inclusion,

$$\mathcal{N}\big(V_D, \eta, \|\cdot\|_\infty\big) \leq \mathcal{N}\big(\mathcal{B}_{R,W}^{\mathrm{BL}}, \eta, \|\cdot\|_\infty\big)$$

for all $\eta > 0$. Taking logarithms yields the last displayed inequality in the lemma. $\square$

### F.2. Proof of Theorem 4.4

*Proof.* The proof is identical in structure to the argument given previously for the generic Sobolev entropy case; we simply replace the abstract entropy assumption by Lemma F.1. We sketch the main steps for completeness.

Let

$$Z := \sup_{g \in V_D} \big|J_\varepsilon(g) - \widehat{J}_{n,\varepsilon}(g)\big|$$

$$\leq \underbrace{\sup_{f \in \Phi_{a,\varepsilon}} \big|(P_m - P)f\big|}_{=:X_m} + \underbrace{\sup_{f \in \Phi_{b,\varepsilon}} \big|(Q_n - Q)f\big|}_{=:Y_n}.$$

where $\Phi_{a,\varepsilon} := \{\phi_{a,\varepsilon}(g) : g \in V_D\}$ and $\Phi_{b,\varepsilon} := \{\phi_{b,\varepsilon}(g) : g \in V_D\}$.

We first Apply a Bousquet/Bernstein-type inequality to each class $\Phi_{p,\varepsilon}$ with envelope $M_{p,\varepsilon}$ yields

$$\Pr(X_m \geq \mathbb{E}X_m + \delta) \leq \exp\big(-c\,m\delta^2/M_{a,\varepsilon}^2\big),$$
$$\Pr(Y_n \geq \mathbb{E}Y_n + \delta) \leq \exp\big(-c\,n\delta^2/M_{b,\varepsilon}^2\big).$$

By a union bound,

$$\Pr\Big(Z \geq (\mathbb{E}X_m + \mathbb{E}Y_n) + \delta\Big) \; \leq \; \exp\Big(-c\,\tfrac{m\delta^2}{M_{a,\varepsilon}^2}\Big)$$
$$+ \exp\Big(-c\,\tfrac{n\delta^2}{M_{b,\varepsilon}^2}\Big).$$

Then we utilize Lipschitz contraction and transfer entropy from $V_D$ to $\Phi_{p,\varepsilon}$. Because $\phi_{p,\varepsilon}$ is globally Lipschitz with constant $L_{p,\varepsilon}$, we have

$$\ln\mathcal{N}\big(\Phi_{p,\varepsilon}, \epsilon, L_2(P)\big) \; \leq \; \ln\mathcal{N}\big(V_D, \epsilon/L_{p,\varepsilon}, \|\cdot\|_\infty\big).$$

Using $V_D \subset \mathcal{B}_{R_\star, W}^{\mathrm{BL}}$ and Lemma F.1,

$$\ln\mathcal{N}\big(\Phi_{p,\varepsilon}, \epsilon, L_2(P)\big)$$
$$\leq \Big(\frac{C_h(W)R_\star\sqrt{S_k}L_{p,\varepsilon}}{\epsilon}\Big)^{2d/h}$$
$$\leq \Big(\frac{A_\varepsilon}{\epsilon}\Big)^{2d/h}.$$

for $p \in \{a, b\}$.

Dudley's entropy integral (via the $\gamma_2$ functional) gives

$$\mathbb{E}X_m \; \lesssim \; \frac{1}{\sqrt{m}} \int_0^{\mathrm{diam}(\Phi_{a,\varepsilon})} \sqrt{\ln\mathcal{N}(\Phi_{a,\varepsilon}, \epsilon, L_2(P))}\, d\epsilon.$$

With the bound $\ln\mathcal{N} \lesssim (A_\varepsilon/\epsilon)^{2d/h}$, the standard fixed-point localization yields

$$\mathbb{E}X_m \; \lesssim \; A_\varepsilon^{\frac{d}{d+h}}\, m^{-\frac{h}{2(d+h)}}, \qquad \mathbb{E}Y_n \; \lesssim \; A_\varepsilon^{\frac{d}{d+h}}\, n^{-\frac{h}{2(d+h)}}.$$

If we choose $h > d$ (possible for any fixed bandlimit $W$ by Lemma F.1), the Dudley integral is integrable at $0$, yielding the simpler parametric-type bounds

$$\mathbb{E}X_m \lesssim A_\varepsilon/\sqrt{m}, \qquad \mathbb{E}Y_n \lesssim A_\varepsilon/\sqrt{n}.$$

Combining Steps 1 and 3 gives the asserted high-probability bound with offset $\mathbb{E}_{n,\varepsilon} = \mathbb{E}X_m + \mathbb{E}Y_n$. Evaluating the supremum at $g = g_{D,R}^\dagger$ yields the desired control of $\mathcal{E}_{\mathrm{stat}}$. $\qquad\square$

# G. Proofs for Section 4.5

This appendix collects the technical assumptions and derivations omitted from the main text in Section 4.5. We first record standing assumptions and derivative identities, then establish intrinsic curvature and the local PL region, prove Theorem 4.5, and finally derive the bridge (28) from $\widehat{F}$-suboptimality to $\mathcal{E}_{\mathrm{opt}}$.

## G.1. Standing assumptions and basic identities

**Assumption G.1** (Norm equivalence between $\mathbb{R}^D$ and $\mathcal{H}_k$). There exist constants $c_D^-, c_D^+ > 0$ such that for all $w \in \mathbb{B}_{R_w}$,

$$c_D^- \|w\|_2 \leq \|f_w\|_{\mathcal{H}_k} \leq c_D^+ \|w\|_2.$$

**Assumption G.2** (Bounded stochastic variance (additive)). The stochastic oracle satisfies $\mathbb{E}[g_t \mid w_t] = \nabla\widehat{F}(w_t)$ and

$$\mathbb{E}\big[\|g_t - \nabla\widehat{F}(w_t)\|_2^2 \mid w_t\big] \leq \sigma^2.$$

With the stochastic oracle defined, we explicitly state the Armijo line-search procedure and update rule used in our analysis.

**Algorithm definition (ArmijoSGD).** For completeness, we specify the update rule and line-search condition referenced in Section 4.5. At step $t$, given the current iterate $w_t$ and a stochastic gradient $g_t$, the algorithm searches for a step size $\eta \in \{\eta_{\max}\beta^k\}_{k=0}^{K_{\max}}$. The Armijo condition requires a sufficient decrease in the current batch objective (denoted loosely as $\widehat{F}_t$):

$$\widehat{F}_t(w_t - \eta g_t) \le \widehat{F}_t(w_t) - c \cdot \eta \|g_t\|_2^2. \tag{53}$$

The algorithm selects the largest $\eta_t$ in the candidate set satisfying (53). If no such step is found after $K_{\max}$ trials, we set $\eta_t = 0$. The iterate is then updated via projection onto the feasible set $\mathbb{B}_{R_w}$:

$$w_{t+1} = \Pi_{\mathbb{B}_{R_w}}\big(w_t - \eta_t g_t\big). \tag{54}$$

**Assumption G.3** (Data regularity for non-degeneracy). *The distribution $Q$ is supported on $\mathcal{M} \subset \mathbb{R}^d$ with non-empty interior and admits a density (is absolutely continuous w.r.t. Lebesgue measure on its support).*

**Lemma G.4** (Strict positive definiteness of feature covariance). *Let $\psi(x) = [\psi_1(x), \ldots, \psi_D(x)]^\top$ with $\psi_j(x) = \sqrt{2}\cos(\omega_j^\top x + b_j)$, frequencies $\omega_j$ sampled i.i.d. from a continuous spectral density and $b_j \sim \mathrm{Unif}[0, 2\pi]$. Under Assumption G.3, for any fixed $D$,*

$$\Sigma_Q := \mathbb{E}_{y \sim Q}\big[\psi(y)\psi(y)^\top\big]$$

*is strictly positive definite with probability 1 (over $\{\omega_j, b_j\}_{j=1}^D$), hence $\nu_0 := \lambda_{\min}(\Sigma_Q) > 0$.*

**Derivative identities (used in curvature and smoothness bounds).** For any $p > 0$ and $\varepsilon > 0$,

$$\begin{aligned}
\phi'_{p,\varepsilon}(u) &= p\, u\, (u^2 + \varepsilon^2)^{\frac{p}{2}-1}, \\
\phi''_{p,\varepsilon}(u) &= p\, (u^2 + \varepsilon^2)^{\frac{p}{2}-2}\big(\varepsilon^2 + (p-1)u^2\big).
\end{aligned} \tag{55}$$

With $s_i(w) := w^\top\psi(x_i)$ and $t_j(w) := w^\top\psi(y_j)$, the gradient and Hessian of $\widehat{F}$ are given by (56)–(57).

$$\begin{aligned}
\nabla\widehat{F}(w) = &-\frac{1}{m}\sum_{i=1}^{m}\phi'_{a,\varepsilon}\big(s_i(w)\big)\,\psi(x_i) \\
&+ \frac{1}{n}\sum_{j=1}^{n}\phi'_{b,\varepsilon}\big(t_j(w)\big)\,\psi(y_j) + \lambda w,
\end{aligned} \tag{56}$$

$$\begin{aligned}
\nabla^2\widehat{F}(w) = &-\frac{1}{m}\sum_{i=1}^{m}\phi''_{a,\varepsilon}\big(s_i(w)\big)\,\psi(x_i)\psi(x_i)^\top \\
&+ \frac{1}{n}\sum_{j=1}^{n}\phi''_{b,\varepsilon}\big(t_j(w)\big)\,\psi(y_j)\psi(y_j)^\top + \lambda I_D.
\end{aligned} \tag{57}$$

## G.2. Proof of Lemma G.5

**Lemma G.5** (Structural and Geometric Properties of the Objective). *Assume $S_k := \sup_{x \in \mathcal{M}} k(x, x) < \infty$ and Assumption G.1. Then for all $w \in \mathbb{B}_{R_w}$,*

$$\|f_w\|_\infty \le \sqrt{S_k}\,\|f_w\|_{\mathcal{H}_k} \le \sqrt{S_k}\,c_D^+\,R_w =: B_{R_w}. \tag{58}$$

*(i) Intrinsic curvature and local PL.* Decompose the Hessian as

$$\nabla^2\widehat{F}(w) = H_{\mathrm{data}}(w) + \lambda I_D, \tag{59}$$

*where*

$$\begin{aligned}
H_{\mathrm{data}}(w) := &-\frac{1}{m}\sum_{i=1}^{m}\phi''_{a,\varepsilon}\big(s_i(w)\big)\,\psi(x_i)\psi(x_i)^\top \\
&+ \frac{1}{n}\sum_{j=1}^{n}\phi''_{b,\varepsilon}\big(t_j(w)\big)\,\psi(y_j)\psi(y_j)^\top.
\end{aligned} \tag{60}$$

*Define the intrinsic (non-degeneracy) margin*

$$\kappa_{\mathrm{nd}} := \inf_{|u| \leq B_{R_w}} \inf_{r \in [c_\ell, c_u]} \left( \phi_{b,\varepsilon}''(u) - r\,\phi_{a,\varepsilon}''(u) \right). \tag{61}$$

*Under Assumption G.3, the* population *data curvature at $w_\lambda^\star$ satisfies*

$$\mathbb{E}\big[ H_{\mathrm{data}}(w_\lambda^\star) \big] \;\succeq\; \kappa_{\mathrm{nd}}\,\Sigma_Q \;\succeq\; \kappa_{\mathrm{nd}}\nu_0\,I_D. \tag{62}$$

*Hence when $\kappa_{\mathrm{nd}} > 0$, the dominant positive curvature is intrinsic (Fisher-information-like), arising from the weighted feature covariance under $Q$; $\lambda$ contributes only through the additive $\lambda I_D$ term in (59). In particular, if the empirical Hessian at $w_\lambda^\star$ is positive definite, define*

$$\mu_{\mathrm{loc}} := \frac{1}{2}\,\lambda_{\min}\big( \nabla^2 \widehat{F}(w_\lambda^\star) \big) > 0. \tag{63}$$

*By continuity of $w \mapsto \nabla^2 \widehat{F}(w)$, there exists $r_\star > 0$ such that on $\mathcal{S}_\star$,*

$$\nabla^2 \widehat{F}(w) \;\succeq\; \mu_{\mathrm{loc}}\,I_D, \qquad \forall w \in \mathcal{S}_\star. \tag{64}$$

*Consequently, $\widehat{F}$ satisfies the local Polyak–Łojasiewicz (PL) inequality on $\mathcal{S}_\star$:*

$$\frac{1}{2}\|\nabla \widehat{F}(w)\|_2^2 \;\geq\; \mu_{\mathrm{loc}}\big( \widehat{F}(w) - \widehat{F}(w_\lambda^\star) \big), \qquad \forall w \in \mathcal{S}_\star. \tag{65}$$

***(ii) Global smoothness on the feasible ball.** Let*

$$\begin{aligned} S_\psi &:= \sup_{z \in \mathcal{M}} \|\psi(z)\|_2^2, \\ \overline{L}_p &:= \sup_{|u| \leq B_{R_w}} \big| \phi_{p,\varepsilon}''(u) \big| \quad (p \in \{a,b\}). \end{aligned}$$

*Then $\widehat{F}$ has an L-Lipschitz gradient on $\mathbb{B}_{R_w}$ with*

$$\begin{aligned} \|\nabla \widehat{F}(u) - \nabla \widehat{F}(v)\|_2 &\leq L\,\|u - v\|_2, \\ \text{where} \quad L &:= \lambda + (\overline{L}_a + \overline{L}_b)S_\psi. \end{aligned} \tag{66}$$

*Proof.* First, to establish the uniform range bound stated in (58), we observe the following properties of the function class. For any $w \in \mathbb{B}_{R_w}$, Assumption G.1 gives $\|f_w\|_{\mathcal{H}_k} \leq c_D^+\|w\|_2 \leq c_D^+ R_w$. Since $k$ is bounded with $S_k = \sup_{x \in \mathcal{M}} k(x,x) < \infty$, the reproducing property yields $\|f_w\|_\infty \leq \sqrt{S_k}\,\|f_w\|_{\mathcal{H}_k} \leq \sqrt{S_k}\,c_D^+ R_w$.

Next, we turn our attention to deriving the population intrinsic curvature bound (62) by analyzing the Hessian at the population level. Let $r(x) := \frac{p(x)}{q(x)}$, so $r(x) \in [c_\ell, c_u]$ by Assumption G.3. At the population level,

$$H_{\mathrm{data}}(w) = -\mathbb{E}_{X \sim P}\big[ \phi_{a,\varepsilon}''(w^\top \psi(X))\,\psi(X)\psi(X)^\top \big] + \mathbb{E}_{Y \sim Q}\big[ \phi_{b,\varepsilon}''(w^\top \psi(Y))\,\psi(Y)\psi(Y)^\top \big].$$

Using $\mathbb{E}_P[\cdot] = \mathbb{E}_Q[r(\cdot)\,\cdot]$, for any $v \in \mathbb{R}^D$,

$$v^\top \mathbb{E}\big[ H_{\mathrm{data}}(w) \big] v = \mathbb{E}_{Y \sim Q}\Big[ \big( \phi_{b,\varepsilon}''(w^\top \psi(Y)) - r(Y)\phi_{a,\varepsilon}''(w^\top \psi(Y)) \big)\,(v^\top \psi(Y))^2 \Big].$$

For every $Y$, $u := w^\top \psi(Y) = f_w(Y)$ satisfies $|u| \leq \|f_w\|_\infty \leq B_{R_w}$, and $r(Y) \in [c_\ell, c_u]$. By (61), $\phi_{b,\varepsilon}''(u) - r(Y)\phi_{a,\varepsilon}''(u) \geq \kappa_{\mathrm{nd}}$. Therefore,

$$v^\top \mathbb{E}\big[ H_{\mathrm{data}}(w) \big] v \geq \kappa_{\mathrm{nd}}\,\mathbb{E}_{Y \sim Q}\big[ (v^\top \psi(Y))^2 \big] = \kappa_{\mathrm{nd}}\,v^\top \Sigma_Q v,$$

hence $\mathbb{E}[H_{\mathrm{data}}(w)] \succeq \kappa_{\mathrm{nd}}\Sigma_Q \succeq \kappa_{\mathrm{nd}}\nu_0 I_D$. Specializing to $w = w_\lambda^\star$ gives (62).

We then proceed by demonstrating the global $L$-smoothness of the objective function on $\mathbb{B}_{R_w}$, which leads to the bound in (66). From (57), for any $w \in \mathbb{B}_{R_w}$,

$$\nabla^2 \widehat{F}(w) = -\frac{1}{m}\sum_{i=1}^{m} \phi''_{a,\varepsilon}(s_i(w))\,\psi(x_i)\psi(x_i)^\top + \frac{1}{n}\sum_{j=1}^{n} \phi''_{b,\varepsilon}(t_j(w))\,\psi(y_j)\psi(y_j)^\top + \lambda I_D.$$

Each rank-one matrix $\psi(z)\psi(z)^\top$ has operator norm $\|\psi(z)\|_2^2 \le S_\psi$. Moreover, $|s_i(w)|, |t_j(w)| \le \|f_w\|_\infty \le B_{R_w}$, so $|\phi''_{p,\varepsilon}(\cdot)| \le \overline{L}_p$ on this range. Thus,

$$\left\|\nabla^2 \widehat{F}(w)\right\|_{\mathrm{op}} \le (\overline{L}_a + \overline{L}_b)S_\psi + \lambda =: L,$$

which implies (66) by the mean-value theorem.

Building on the smoothness properties, we now establish the existence of a local strong-convexity neighborhood as described in (64). Assume $\nabla^2 \widehat{F}(w_\lambda^\star)$ is positive definite and define $\mu_{\mathrm{loc}} := \frac{1}{2}\lambda_{\min}(\nabla^2 \widehat{F}(w_\lambda^\star)) > 0$. Since $\varepsilon > 0$, $\phi_{p,\varepsilon}$ is smooth and $\phi'''_{p,\varepsilon}$ is bounded on $[-B_{R_w}, B_{R_w}]$; let $\overline{M}_p := \sup_{|u| \le B_{R_w}} |\phi'''_{p,\varepsilon}(u)| < \infty$. Then $w \mapsto \nabla^2 \widehat{F}(w)$ is locally Lipschitz in operator norm on $\mathbb{B}_{R_w}$, so there exists $r_\star > 0$ such that for all $\|w - w_\lambda^\star\|_2 \le r_\star$, $\|\nabla^2 \widehat{F}(w) - \nabla^2 \widehat{F}(w_\lambda^\star)\|_{\mathrm{op}} \le \mu_{\mathrm{loc}}$. Hence $\nabla^2 \widehat{F}(w) \succeq \mu_{\mathrm{loc}} I_D$ on $\mathcal{S}_\star$, yielding (64).

Finally, the local strong convexity established above directly implies the local PL inequality (65) as follows. On $\mathcal{S}_\star$, (64) implies $\widehat{F}$ is $\mu_{\mathrm{loc}}$-strongly convex, so for $u = w_\lambda^\star$,

$$\widehat{F}(w) - \widehat{F}(w_\lambda^\star) \le \langle \nabla \widehat{F}(w), w - w_\lambda^\star \rangle - \frac{\mu_{\mathrm{loc}}}{2}\|w - w_\lambda^\star\|_2^2 \le \frac{1}{2\mu_{\mathrm{loc}}}\|\nabla \widehat{F}(w)\|_2^2,$$

which is equivalent to (65). $\qquad\square$

### G.3. Proof of Theorem 4.5

*Proof.* Define $\Delta_F(t) = \widehat{F}(w_{t \wedge \tau}) - \widehat{F}(w_\lambda^\star)$. Fix $t$ and condition on $w_t$.

We begin the proof by deriving a one-step inequality resulting from $L$-smoothness and the bounded variance assumption. When $t < \tau$, we have $w_t \in \mathcal{S}_\star \subset \mathbb{B}_{R_w}$. By $L$-smoothness of $\widehat{F}$ on $\mathbb{B}_{R_w}$, for the (unprojected) tentative step $u_{t+1} = w_t - \eta_t g_t$ one has

$$\widehat{F}(u_{t+1}) \le \widehat{F}(w_t) - \eta_t \langle \nabla \widehat{F}(w_t), g_t \rangle + \frac{L\eta_t^2}{2}\|g_t\|_2^2.$$

Since $w_{t+1} = \Pi_{\mathbb{B}_{R_w}}(u_{t+1})$ and $\Pi_{\mathbb{B}_{R_w}}$ is non-expansive, the same inequality applies with $w_{t+1}$ in place of $u_{t+1}$ (equivalently, use the standard smoothness bound along the projected step). Write $g_t = \nabla \widehat{F}(w_t) + \xi_t$ with $\mathbb{E}[\xi_t \mid w_t] = 0$ and $\mathbb{E}[\|\xi_t\|_2^2 \mid w_t] \le \sigma^2$ (Assumption G.2). Then

$$\mathbb{E}\big[\langle \nabla \widehat{F}(w_t), g_t \rangle \mid w_t\big] = \|\nabla \widehat{F}(w_t)\|_2^2, \qquad \mathbb{E}\big[\|g_t\|_2^2 \mid w_t\big] \le \|\nabla \widehat{F}(w_t)\|_2^2 + \sigma^2.$$

Taking conditional expectation gives, on $\{t < \tau\}$,

$$\mathbb{E}\big[\widehat{F}(w_{t+1}) \mid w_t\big] \le \widehat{F}(w_t) - \eta_t\Big(1 - \frac{L\eta_t}{2}\Big)\|\nabla \widehat{F}(w_t)\|_2^2 + \frac{L\eta_t^2}{2}\sigma^2.$$

Since $\eta_t \le \eta_{\max} \le 1/(2L)$, we have $1 - \frac{L\eta_t}{2} \ge \frac{3}{4}$, hence

$$\mathbb{E}\big[\widehat{F}(w_{t+1}) - \widehat{F}(w_\lambda^\star) \mid w_t\big] \le \big(\widehat{F}(w_t) - \widehat{F}(w_\lambda^\star)\big) - \frac{\eta_t}{2}\|\nabla \widehat{F}(w_t)\|_2^2 + \frac{L\eta_t^2}{2}\sigma^2, \qquad (t < \tau). \qquad (67)$$

Next, we incorporate the local PL inequality into the previous bound and account for the stopping condition defined by $\tau$. When $t < \tau$, $w_t \in \mathcal{S}_\star$ and (65) yields $\frac{1}{2}\|\nabla \widehat{F}(w_t)\|_2^2 \ge \mu_{\mathrm{loc}}\big(\widehat{F}(w_t) - \widehat{F}(w_\lambda^\star)\big)$. Plugging into (67) gives

$$\mathbb{E}\big[\widehat{F}(w_{t+1}) - \widehat{F}(w_\lambda^\star) \mid w_t\big] \le (1 - \eta_t\mu_{\mathrm{loc}})\big(\widehat{F}(w_t) - \widehat{F}(w_\lambda^\star)\big) + \frac{L\eta_t^2}{2}\sigma^2, \qquad (t < \tau).$$

For $t \geq \tau$, $\Delta_F(t+1) = \Delta_F(t)$ by definition of the stopped process. Combining both cases yields

$$\mathbb{E}\big[\Delta_F(t+1)\big] \leq \mathbb{E}\big[(1 - \eta_t \mu_{\text{loc}})\Delta_F(t)\big] + \frac{L}{2}\mathbb{E}\big[\eta_t^2 \mathbf{1}_{\{t < \tau\}}\big]\sigma^2.$$

Under the theorem assumption that whenever $t < \tau$ the backtracking accepts a non-null step, $\eta_t \geq \eta_{\min}$ on $\{t < \tau\}$, and always $\eta_t \leq \eta_{\max}$. Hence

$$\mathbb{E}\big[\Delta_F(t+1)\big] \leq (1 - \eta_{\min}\mu_{\text{loc}})\,\mathbb{E}\big[\Delta_F(t)\big] + \frac{L\eta_{\max}^2}{2}\sigma^2.$$

Finally, we unroll the resulting affine recursion to obtain the convergence rate. Let $a_t := \mathbb{E}[\Delta_F(t)]$, $q := \eta_{\min}\mu_{\text{loc}}$ and $b := \frac{L\eta_{\max}^2}{2}\sigma^2$. Then $a_{t+1} \leq (1-q)a_t + b$, so

$$a_t \leq (1-q)^t a_0 + \frac{b}{q} = (1 - \eta_{\min}\mu_{\text{loc}})^t \big(\widehat{F}(w_0) - \widehat{F}(w_\lambda^\star)\big) + \frac{L\eta_{\max}^2}{2\,\eta_{\min}\mu_{\text{loc}}}\,\sigma^2,$$

which is (27). $\qquad\square$

# H. Characterization of $\alpha$-Divergence

The Amari $\alpha$-divergence uniquely belongs to both the $f$-divergence and Bregman divergence classes, making it the canonical divergence for dually flat positive measure spaces (Amari, 2009). This dual membership has important geometric and computational implications.

The $\phi$-divergence class is defined by information monotonicity—the property that coarse-graining cannot increase divergence. For a convex function $\phi(u)$ with $\phi(1) = 0$, the general form is $D_\phi(P\|Q) = \int q(x)\phi(p(x)/q(x))dx$.

Consistent with the normalization in (1), the $\alpha$-divergence uses the generator $\phi_\alpha(u) = \frac{1}{\alpha(1-\alpha)}(1 - u^\alpha)$, inheriting information monotonicity and inducing the Fisher metric with $\alpha$-affine connections

In contrast, Bregman divergences are characterized by dual flatness. For a convex potential $\beta(r)$, the Bregman divergence is $D_\beta(r\|s) = \beta(r) - \beta(s) - \langle\nabla\beta(s), r - s\rangle$. While these divergences create dually flat geometries, they typically lack information monotonicity and do not induce the Fisher metric.

The $\alpha$-divergence achieves dual membership through a coordinate transformation in positive measure space $M$. Using $\alpha$-coordinates $r_i^\alpha = m_i^{(1-\alpha)/2}$ with potential $U_\alpha(r) = \frac{2}{1-\alpha}\sum_i (r_i^\alpha)^{(1+\alpha)/(1-\alpha)}$, the Bregman divergence in these coordinates exactly recovers the $\alpha$-divergence. This construction reveals why $\alpha$-divergences are canonical: they uniquely satisfy both information monotonicity and dual flatness simultaneously.

This dual nature provides both robustness and computational efficiency. Information monotonicity ensures stability under marginalization with the Fisher metric as the natural geometry, while the Bregman structure enables generalized Pythagorean and projection theorems that simplify optimization.

# I. $\alpha$-Divergence Applications in Machine Learning

The theoretical properties of Amari $\alpha$-divergence have motivated diverse applications across generative modeling, where flexible trade-offs between distribution matching objectives are beneficial (Daudel et al., 2023a; Li, 2025). In generative adversarial networks, the Alpha-GAN framework employs $\alpha$-divergence as the training objective, generalizing the standard KL or Jensen-Shannon divergence minimization to control emphasis on real versus generated distributions (Cai et al., 2020). By introducing hyperparameters that balance forward and reverse divergence properties, Alpha-GAN achieves improved training stability and reduced mode collapse, with experimental results demonstrating competitive or superior Fréchet Inception Distance scores compared to WGAN and WGAN-GP. Recent theoretical analyses in variational inference have rigorously characterized the asymptotic behavior of $\alpha$-divergence minimization, formalizing how the choice of divergence parameter explicitly controls the trade-off between mode-seeking behavior and mass coverage within exponential families (Daudel et al., 2023a; Zhang et al., 2019a). Additionally, transport $\alpha$-divergences provide a one-parameter family interpolating between transport metrics in Wasserstein-2 space, with applications to flow matching and normalizing flows (Li, 2025), demonstrating that $\alpha$-divergence principles transfer naturally to continuous-time generative models beyond traditional GANs.

Beyond generative modeling, $\alpha$-divergence has found applications across reinforcement learning, variational inference, and knowledge distillation (Koyamada et al., 2017; Geffner & Domke, 2021; Villacampa-Calvo et al., 2022; Daudel et al., 2023b; Shin et al., 2025; Wang et al., 2025). In reinforcement learning, $\alpha$-divergence bridges maximum likelihood and policy gradient methods, with $\alpha \to 0$ corresponding to ML and $\alpha \to 1$ to RL, enabling improved performance on sequence generation tasks such as machine translation (Koyamada et al., 2017), while alphaPPO replaces KL-divergence constraints with $\alpha$-divergence for enhanced policy update stability (Xu et al., 2023). For variational inference, different $\alpha$ values control mode-covering versus mode-seeking behavior, though unbiased minimization faces challenges in high dimensions due to exponentially degrading gradient estimator signal-to-noise ratios (Geffner & Domke, 2021). Monotonic $\alpha$-divergence minimization algorithms with convergence guarantees have been developed for deep Gaussian processes, improving inference accuracy over standard variational approaches (Daudel et al., 2023b; Villacampa-Calvo et al., 2022). In knowledge distillation, the ABKD framework uses $\alpha$-$\beta$-divergence to balance hardness and confidence concentration (Wang et al., 2025), while the AMiD framework introduces $\alpha$-mixture assistant distributions that provide continuous control over interpolation geometry between teacher and student models, modulating mode-covering and mode-seeking during distillation for improved stability and performance (Shin et al., 2025). Additional applications include model integration (Amari, 2007), generative model evaluation through divergence frontiers (Djolonga et al., 2020), and entropy production estimation in statistical physics (Kwon & Baek, 2024).

## J. Hyperparameter Schedules and Implementation Details

This appendix provides the full pseudocode of the improved estimator and the derivations of the hyperparameter schedules used in Section 5. Throughout we write $N := m \wedge n$ and use the RFF map $\psi$ in (3).

### J.1. Pseudocode

---

**Algorithm 1** Bandwidth-limited RFF estimator with intrinsic-geometry optimization

---

1: **Input:** samples $\{x_i\}_{i=1}^m \sim P$, $\{y_j\}_{j=1}^n \sim Q$, kernel $k$, exponents $a, b$, budget parameters $(K_{\max}, \beta, c)$, batch size $B$.
2: Set $N \leftarrow m \wedge n$.
3: **Calibration (spectral coupling and budgets).**
4: Choose RKHS radius $R = R(N)$ and bandlimit $W = W(R)$ as in Theorem 4.3 (one admissible balancing rule is recorded in Appendix J.2).
5: Sample $\omega_{1:D} \overset{\text{i.i.d.}}{\sim} \Lambda_W$ and $b_{1:D} \overset{\text{i.i.d.}}{\sim} \text{Unif}[0, 2\pi]$, construct $\psi(\cdot)$ by (3).
6: Set smoothing $\varepsilon \asymp N^{-1/2}$.
7: Set feature budget

$$D \gtrsim d\, R^2\, W^2\, N$$

(up to logarithmic factors; see Appendix J.2).

8: Choose $\lambda$ such that $\lambda R_w^2 = o(N^{-1/2})$ (e.g. $\lambda \asymp N^{-1} R_w^{-2}$).
9: **Optimization (ArmijoSGD on $\widehat{F}$).**
10: Initialize $w_0 = 0$ and set $\eta_{\max} \leq (2L)^{-1}$, $\eta_{\min} = \eta_{\max} \beta^{K_{\max}}$.
11: **for** $t = 0, 1, 2, \ldots, T - 1$ **do**
12:     Draw mini-batches $\mathcal{B}_t^P \subset \{x_i\}$, $\mathcal{B}_t^Q \subset \{y_j\}$ of size $B$.
13:     Form stochastic gradient $g_t$ of $\widehat{F}$ at $w_t$ using $(\mathcal{B}_t^P, \mathcal{B}_t^Q)$.
14:     Backtrack over $\eta \in \{\eta_{\max} \beta^k\}_{k=0}^{K_{\max}}$ until Armijo decrease holds:

$$\widehat{F}\Big(\Pi_{\mathbb{B}_{R_w}}(w_t - \eta g_t)\Big) \leq \widehat{F}(w_t) - c\, \eta\, \|g_t\|_2^2.$$

15:     Update $w_{t+1} \leftarrow \Pi_{\mathbb{B}_{R_w}}(w_t - \eta_t g_t)$.
16: **end for**
17: **Output:** critic $\widehat{g}_{\varepsilon,T}(x) = w_T^\top \psi(x)$ and the plug-in estimator defined in Section 4 (using $\widehat{g}_{\varepsilon,T}$).

---

### J.2. Deriving the schedules

We record the constraints used to choose $(\varepsilon, D, \lambda)$ and one convenient balancing rule for $(R, W)$.

**Smoothing level.**  The proxy analysis yields $|\mathcal{E}_{\mathrm{proxy}}| \lesssim \varepsilon^a + \varepsilon^b$. Setting $\varepsilon \asymp N^{-1/2}$ gives $\varepsilon^a + \varepsilon^b \asymp N^{-a/2} + N^{-b/2}$, so the proxy bias does not exceed the statistical scale in the bandlimited regime.

**Feature budget.**  Section 5.3 (bandlimited RFF approximation) gives, with high probability,

$$|\mathcal{E}_{\mathrm{feat}}| \;\lesssim\; L_{\varepsilon,B}\, R\left( W\sqrt{\frac{d}{D}} + \sqrt{\frac{\log(1/\delta)}{D}} \right).$$

To keep $|\mathcal{E}_{\mathrm{feat}}| \lesssim N^{-1/2}$, it suffices (up to logarithmic factors) to impose

$$R\,W\sqrt{\frac{d}{D}} \;\lesssim\; N^{-1/2}, \qquad \text{hence} \qquad D \;\gtrsim\; d\,R^2\,W^2\,N.$$

This is the missing $N$-factor in the naive rule $D \gtrsim dR^2 W^2$.

**Regularization level.**  The telescoping bridge in (28) contains the additive term $\frac{\lambda}{2}R_w^2$. To ensure regularization does not dominate the statistical scale, we enforce $\lambda R_w^2 = o(N^{-1/2})$. This choice is motivated purely by bias control; the stability and local linear rate are governed by intrinsic geometry through $\mu_{\mathrm{loc}}$ (Section 4.5), not by inflating $\lambda$.

**Choosing $(R, W)$.**  The approximation analysis provides an intrinsic bias term that decreases with $R$ (e.g. $\|f^\star - g_R\|_\infty \lesssim R^{-\gamma}$ for some $\gamma > 0$) and a coupling $W = W(R)$ from the bandlimited proxy construction. A convenient balancing rule is to choose $R = R(N)$ so that

$$R^{-\gamma} \;\asymp\; N^{-1/2},$$

and then set $W = W(R)$ according to the spectral-window mapping in Theorem 4.3. This ensures that the intrinsic bias and the bandlimited statistical scale match, while $D$ is then chosen by the feature-budget condition above.

### J.3. Operational notes (spectral truncation and projection)

Truncating the spectrum enforces the bandlimit used throughout Section 5, so the estimator inherits the same conditioning window and entropy control. Projecting onto $\mathbb{B}_{R_w}$ keeps iterates within the range where the local geometry (PL region) and the uniform bounds invoked in the error decomposition remain valid. Armijo backtracking adapts the step size to the local smoothness of the smoothed objective and avoids manual tuning when $\varepsilon$ is small.

## K. Detailed Experimental Setup

### K.1. Experimental Setup of Estimation

We investigate the estimation of the Amari $\alpha$-divergence $D_\alpha^{(A)}(P\|Q)$ between two known distributions, focusing on verifying the theoretical scaling laws with respect to sample size and dimension. To ensure a controlled environment where the density ratio is bounded yet non-trivial, we construct the domain $\mathcal{X} \subset \mathbb{R}^d$ as a compact union of hyper-rectangles. Specifically, for the two-dimensional case ($d = 2$), the support is explicitly defined as $\mathcal{X} = [0.1, 2.0] \times [-1.0, 0.0]$. For the five-dimensional setting ($d = 5$), we extend this construction to a specific product of intervals defined by $[0.1, 2.0] \times [-1.0, 0.0] \times [-2.0, -1.5] \times [-1.0, 1.0] \times [a_5, b_5]$, ensuring a complex support structure. The ten-dimensional case ($d = 10$) is subsequently constructed as the Cartesian product of two such 5-dimensional blocks. Within this domain, the target distribution $P$ is defined as a standard multivariate Gaussian $\mathcal{N}(0, I_d)$ truncated to $\mathcal{X}$, while the reference distribution $Q$ is set to be uniform over the same support, i.e., $Q = \mathrm{Unif}(\mathcal{X})$. The ground truth divergence $D_{\mathrm{true}}$ is computed using a high-precision Monte Carlo estimator utilizing importance sampling with $N_{\mathrm{MC}} = 2 \times 10^5$ samples drawn from the reference distribution.

To validate our approximation theoretic claims, we compare four distinct estimator architectures. The first, denoted as **Basic(NN)**, serves as a standard neural baseline and consists of a Multi-Layer Perceptron (MLP) with a fixed depth of $L = 2$ and a constant width of $H = 256$, utilizing hyperbolic tangent ($\tanh$) activations. The second model, **Adaptive(NN)**, is explicitly designed to test the impact of width scaling on convergence rates. Motivated by our theoretical analysis for Hölder function classes, we evaluate this model under two distinct scaling regimes: an *optimal scaling* where the network width $k_n$ grows proportional to $n^{0.5}$, and a *conservative scaling* where $k_n \propto n^{0.2}$. The third approach is the **RFF-Estimator**. To

bridge theory and practice, we implement the critic using standard Gaussian sampling, which acts as an efficient proxy for the theoretically analyzed truncated estimator since the Gaussian mass outside our effective bandwidth $W(R)$ is negligible ($< 1\%$). Finally, we evaluate **RKHS(NN)**, which is a neural network estimator explicitly regularized to constrain the learned function within an RKHS. Unlike the RFF estimator which uses a fixed basis, RKHS-Net optimizes its weights but enforces a smoothness constraint (via spectral norm regularization or weight penalties) to mimic the kernel space geometry, thus serving as a regularized neural baseline.

Our experimental protocol involves varying the sample size $n$ across the set $\{10^5, \ldots, 5 \times 10^6\}$ and the dimension $d \in \{2, 5, 10\}$. All estimators are trained by minimizing the regularized smoothed objective $\hat{F}(w) = -J_\varepsilon(f_w) + \frac{\lambda}{2}\|w\|^2$. We employ either the Adam optimizer or a custom Armijo-backtracking line search SGD, depending on the stability requirements of the specific estimator. To adhere to the theoretical derivation, the smoothing parameter $\varepsilon_n$ and the regularization coefficient $\lambda_n$ are decayed at rates proportional to $n^{-1/2}$. Performance is strictly assessed using the relative effective error, defined as R-RMSE $= \frac{\|\hat{D} - D_{\text{true}}\|_2}{\|D_{\text{true}}\|_2}$.

## K.2. Detailed Setting of Spatial RFF GAN

We apply our framework to the task of learning high-dimensional manifolds via Generative Adversarial Networks (GANs). While standard approaches often focus on generator modifications, our experimental design introduces a **Spatial RFF Discriminator**, which explicitly incorporates inductive biases via fixed spatial embeddings. The models are trained on the CelebA dataset, resized to a resolution of $64 \times 64$ with pixel values normalized to $[-1, 1]$.

### MODEL ARCHITECTURES

The generator adopts a stable architecture designed to avoid checkerboard artifacts. It initializes from a latent code $z \sim \mathcal{N}(0, I)$ projected to a $4 \times 4 \times 512$ feature map. Feature resolution is progressively increased through a series of robust upsampling blocks, each consisting of nearest-neighbor interpolation, a $3 \times 3$ convolution, Batch Normalization, and LeakyReLU activation. The network outputs a $64 \times 64$ image via a final Tanh activation.

The core innovation lies in the discriminator. It begins with a standard CNN backbone utilizing Spectral Normalization and LeakyReLU, which downsamples the input image to a $4 \times 4$ bottleneck feature map. At this low-resolution stage, we inject deterministic position information via a fixed **Spatial RFF** map. Let $p \in [-1, 1]^2$ denote the normalized coordinates of the $4 \times 4$ grid. The embedding is defined as:

$$v(p) = [\cos(2\pi \mathbf{B}p), \sin(2\pi \mathbf{B}p)], \quad \text{where } \mathbf{B} \sim \mathcal{N}(0, \sigma^2 I). \tag{68}$$

These RFF features are concatenated with the learnable CNN features along the channel dimension. A crucial implementation detail is the **Zero-Initialization** strategy applied to the subsequent $1 \times 1$ fusion convolution: weights corresponding to the RFF channels are initialized to zero, while CNN weights follow standard initialization. This ensures that the discriminator behaves identically to a standard CNN at the start of training, gradually incorporating spatial inductive biases as training progresses. The fused representation is then processed by a $3 \times 3$ spectral-normalized mixing layer before the final scalar scoring.

### LOSS FUNCTION AND OPTIMIZATION

Training is performed by minimizing the smoothed Amari $\alpha$-divergence. We define the smoothed convex function $\varphi_{p,\varepsilon} : \mathbb{R} \to \mathbb{R}$ as:

$$\varphi_{p,\varepsilon}(t) = (t^2 + \varepsilon^2)^{p/2}, \tag{69}$$

where $\varepsilon$ is a smoothing factor. The empirical objective function $J_\varepsilon$ for the discriminator (and adversarially for the generator) is formulated as:

$$\mathcal{L}(D, G) = \mathbb{E}_{x \sim p_{data}} \left[\varphi_{a,\varepsilon}(D(x))\right] - \mathbb{E}_{z \sim p_z} \left[\varphi_{b,\varepsilon}(D(G(z)))\right], \tag{70}$$

where $p_{data}$ is the real data distribution and $p_z$ is the latent prior. In our experiments, we set the shape parameters to $a = 1.5$ and $b = 2.0$, with a smoothing factor $\varepsilon = 10^{-4}$.

To rigorously control the optimization trajectory, we replace the standard Adam optimizer with a custom **Armijo-SGD** optimizer. This optimizer implements a backtracking line search condition, updating parameters only when the sufficient decrease condition $f(w_k - \eta\nabla f(w_k)) \leq f(w_k) - c \cdot \eta\|\nabla f(w_k)\|^2$ is met. We fix the initial step sizes for both networks at

$\eta_G = \eta_D = 10^{-4}$, set the backtracking reduction factor to $\beta = 0.5$, and use a control parameter of $c = 10^{-4}$. We impose $L_2$ regularization on the discriminator weights with $\lambda = 10^{-3}$ as a proxy for RKHS norm control, notably avoiding Gradient Penalty (GP) to demonstrate the inherent stability of our objective.

To comprehensively assess generative quality, we combine distributional feature metrics such as FID and KID for semantic realism (Bińkowski et al., 2018), multi-scale sliced Wasserstein distance for texture consistency(Karras et al., 2018), and frequency-domain statistics inspired by focal-frequency analyses to quantify high-frequency spectral fidelity. All metrics are computed on 50,000 generated samples compared against the training distribution.

### K.3. Custom Optimization Algorithm (Adam + Armijo)

We use Adam to form an adaptive preconditioned direction and then apply Armijo backtracking along that direction to ensure sufficient decrease of the (mini-batch) objective. Algorithm 2 states the minimization-form update used in our GAN experiments.

---

**Algorithm 2** Adam with Armijo backtracking (minimization form)

---

**Require:** Current parameters $\theta$, closure returning $f(\theta)$ and $\nabla f(\theta)$, $\eta_{\text{init}}$, $(\beta_1, \beta_2)$, $\epsilon$, Armijo parameters $(c, \beta)$, max backtracks $B$, minimum step $\eta_{\min}$.
1: Evaluate $f(\theta)$ and gradient $g = \nabla f(\theta)$.
2: Update Adam moments: $m \leftarrow \beta_1 m + (1 - \beta_1)g$, $v \leftarrow \beta_2 v + (1 - \beta_2)g \odot g$.
3: Bias-correct: $\hat{m} \leftarrow m/(1 - \beta_1^t)$, $\hat{v} \leftarrow v/(1 - \beta_2^t)$.
4: Direction (preconditioned): $d \leftarrow \hat{m}/(\sqrt{\hat{v}} + \epsilon)$.
5: Set $\eta \leftarrow \eta_{\text{init}}$ and save $\theta_0 \leftarrow \theta$.
6: **for** $b = 0, 1, \ldots, B$ **do**
7:     Propose $\theta \leftarrow \theta_0 - \eta d$ and evaluate $f(\theta)$ via closure.
8:     **if** $f(\theta) \leq f(\theta_0) - c\,\eta\,\langle g, d \rangle$ **then**
9:         **return** accepted update $(\theta, \eta, b)$.
10:     **end if**
11:     Backtrack: $\eta \leftarrow \beta\eta$.
12:     **if** $\eta < \eta_{\min}$ **then**
13:         **break**
14:     **end if**
15: **end for**
16: Revert $\theta \leftarrow \theta_0$ and **return** no-update.

---

# L. Additional Results

In this section, we analyze the optimization dynamics of different estimators as visualized in Figure 2.

**Optimization Challenges in Neural Baselines.** As shown in the convergence curves, both **Basic (NN)** and **Adaptive (NN)** fail to approximate the ground truth divergence value, plateauing at a significantly lower lower-bound. Despite using the Adam optimizer with standard hyperparameter tuning, these neural network-based variational estimators struggle to minimize the optimization error. We attribute this to the non-convex and ill-conditioned loss landscape typical of deep neural networks in variational divergence estimation. The optimizer likely gets trapped in poor local optima or saddle points, preventing the estimator from reaching the true supremum.

**Geometric Superiority of RFF and RKHS Methods.** In contrast, the **RFF** and **RKHS (NN)** estimators demonstrate superior convergence properties, effectively matching the ground truth. This performance gap is not merely due to expressivity, but rather *optimization stability*. The RFF method transforms the problem into a linear regression in a mapped feature space, resulting in a convex (or near-convex) optimization objective with respect to the weights. Similarly, the RKHS formulation leverages the kernel trick to operate in a theoretically well-behaved functional space. These favorable geometric properties allow standard gradient-based optimizers to efficiently locate the global optimum, thereby reducing the optimization gap that plagues pure neural network approaches.

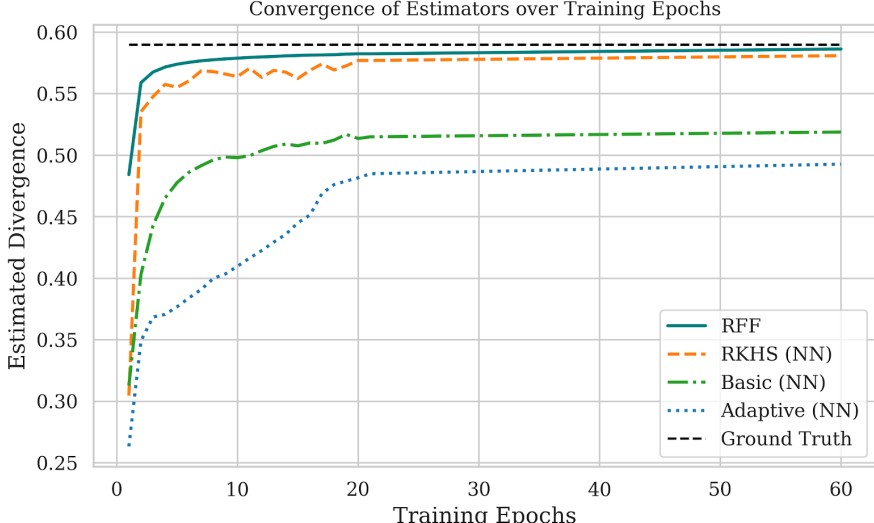

*Figure 2.* **Convergence Analysis of Variational Estimators.** Trajectories of estimated divergence values over training epochs. The **RFF** (teal) and **RKHS (NN)** (orange) estimators rapidly converge toward the ground truth (dashed black line). In contrast, neural network-based baselines (**Basic (NN)** and **Adaptive (NN)**) plateau significantly below the true value, indicating optimization difficulties.

**Impact of Spatial Inductive Bias on Optimization Stability** To verify the theoretical hypothesis that incorporating high-frequency geometry stabilizes the critic, we compare the training dynamics of the unaugmented baseline ($\sigma = 0$) against the Spatial-RFF estimator ($\sigma = 1$).

Figure 4 (Left) illustrates the discriminator loss trajectories. The baseline model exhibits a rapid initial drop followed by a potential plateau, suggesting that the standard CNN discriminator may be overfitting to low-frequency features early in training. In contrast, the RFF-augmented model ($\sigma = 1$) maintains a more sustained learning signal.

Crucially, Figure 4 (Right) reveals the mechanism behind this stability: the gradient norms. The baseline ($\sigma = 0$) suffers from diminishing gradient magnitudes as training progresses, a common pathology in GANs that hampers the generator's ability to learn fine details. However, simply injecting Spatial RFFs ($\sigma = 1$) effectively counters this trend, maintaining significantly healthier gradient norms throughout the optimization process. This confirms that the explicit introduction of spatial inductive bias prevents gradient collapse for high-frequency components, providing the generator with robust feedback to refine spectral details.

**Qualitative Analysis on CelebA.** Figure 3 presents a visual ablation study across different Spatial-RFF bandwidths $\sigma \in \{0, 1, 2, 5\}$. As observed in Figure 3a, the baseline samples ($\sigma = 0$) suffer from severe spectral bias, manifesting as blurry textures and structural incoherence. The lack of high-frequency guidance results in "washed-out" faces where fine details like hair strands and skin texture are absent. As the bandwidth increases to $\sigma = 1$ and $\sigma = 2$ (Figures 3b and 3c), we observe a progressive restoration of geometric integrity. Crucially, the configuration with $\sigma = 5$ (Figure 3d) yields the most perceptually convincing results, producing sharp facial features and realistic high-frequency details. This visual progression corroborates our theoretical hypothesis: enlarging the spectral window of the discriminator via Spatial RFFs effectively mitigates the vanishing gradient problem for high-frequency components, thereby forcing the generator to synthesize fine-grained structures that are otherwise lost in standard CNN-based architectures.

## M. High-Resolution Image Generation and Extended Benchmark Comparisons

In this section, we further examine the scalability of the proposed generative framework on a more complex image distribution by extending the experiment to the **FFHQ 256×256** dataset. The purpose of this study is to evaluate whether the critic-side spectral regularization remains effective at higher resolution and whether it helps reduce the training instabilities and visual artifacts observed in simpler high-frequency ablations. We therefore keep the focus on the same mechanism studied in the main paper: a controlled spectral and modulation pathway added to a modern adversarial backbone, rather than a complete redesign of the generator or a claim of universal high-resolution image-generation superiority. The experiment

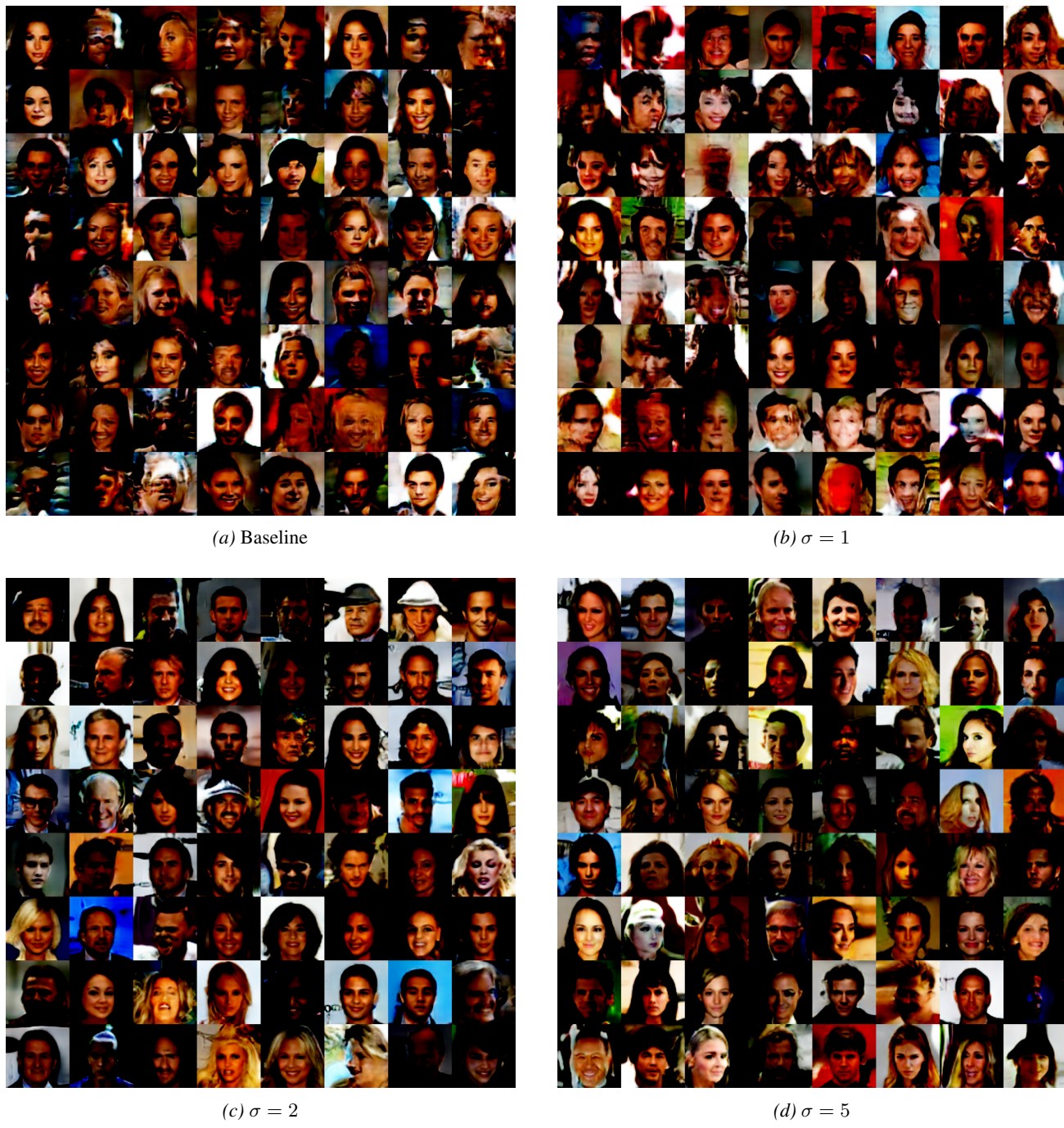

*(a)* Baseline

*(b)* $\sigma = 1$

*(c)* $\sigma = 2$

*(d)* $\sigma = 5$

*Figure 3.* **Qualitative results on CelebA.** Visual comparison of generated samples showing the effect of varying Spatial-RFF bandwidth $\sigma$. The visual quality aligns with the quantitative metrics in Table 1.

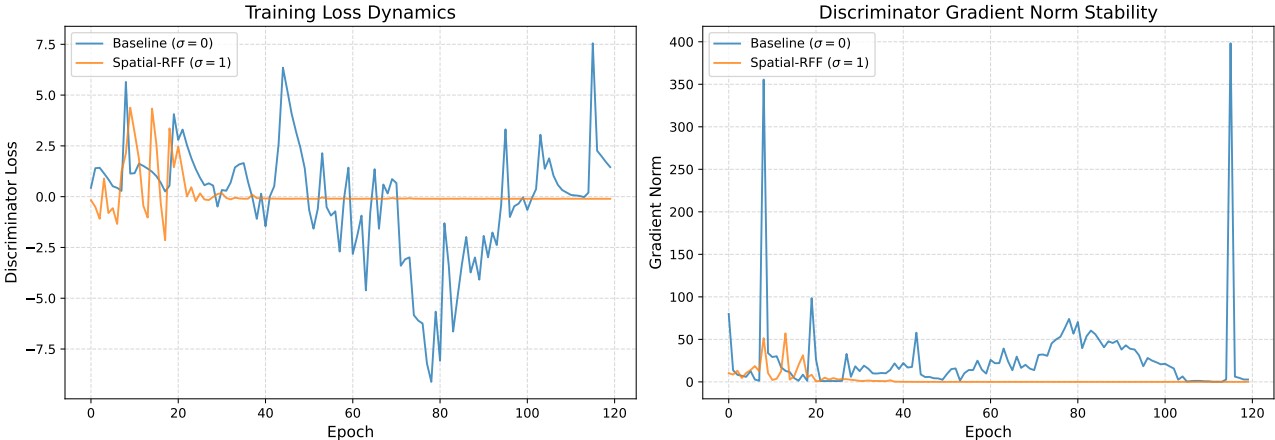

*Figure 4.* **Effect of Spatial Inductive Bias.** Comparison of training dynamics between the Baseline ($\sigma = 0$) and Spatial-RFF ($\sigma = 1$). The introduction of RFF features prevents the decay of gradient norms (Right), ensuring consistent feedback for learning high-frequency details.

uses R3GAN as the base model and reports the unbiased Kernel Inception Distance (KID) as the main quantitative criterion.

To support controllable attribute synthesis while preserving the high-frequency priors of a pre-trained generator, such as skin texture and hair detail, we introduce an **Attribute-Recursive Adaptation (ARA)** module. Rather than using a single-pass label embedding as in a standard conditional GAN, ARA acts as a lightweight feedback path through the synthesis network. During the forward pass, it reads intermediate feature maps, extracts attribute-related representations, and forms a recursive signal that measures how the current synthesis state differs from the target attribute condition. The resulting signal is converted into small modulation vectors, including bias and scale adjustments, with approximately $\sim$189k additional parameters. These modulation vectors do not replace the backbone features directly. Instead, they are added residually to selected affine transformation layers, synthesis biases, and ToRGB layers of the frozen generator, allowing the conditional guidance to move the latent trajectory within the existing data manifold while reducing the risk of injecting out-of-distribution high-frequency noise.

We combine ARA with a Structured Modulation Fine-Tuning (SM-FT) strategy to adapt the high-resolution R3GAN backbone under a limited training budget. In the fine-tuning stage, $99\%$ of the core convolutional kernels in the pre-trained baseline are frozen, and the trainable parameters are restricted to the ARA module, the full mapping network, and the selected synthesis bias and ToRGB modulation parameters. Keeping the mapping network trainable is important in this setting. In early trials, updating only isolated bias terms tended to create structural decoupling, such as floating facial components, because the conditional signal could not be translated into a coherent global style code $w$. Allowing the mapping network to adjust gives the conditional signal a smoother path through the latent representation while still preserving most of the pre-trained generator.

Optimization is performed with Adam. To avoid discriminator dominance in this constrained adaptation regime, we use separate learning rates for the two players, setting the generator learning rate to $\eta_G = 5 \times 10^{-5}$ and the discriminator learning rate to $\eta_D = 2 \times 10^{-5}$. The attribute-recursive scale $\alpha$ is kept in the range 0.01–0.02, which we found sufficient for attribute guidance while avoiding the bias saturation that can otherwise appear as high-saturation color artifacts. The $R_1$ gradient penalty is evaluated every 16 steps, with $\gamma$ adjusted to maintain discriminator smoothness. Final metrics are computed using exponential moving average (EMA) weights collected over the last 200 kimg, which reduces sampling noise in the reported results.

To quantify fidelity and mode coverage at $256 \times 256$ resolution, we report the **Kernel Inception Distance (KID)**. KID is an unbiased estimator based on squared maximum mean discrepancy in Inception feature space, and is therefore useful for measuring distributional mismatch without the sample-size bias associated with FID. We compare against modern GAN and diffusion baselines, including ADM(Dhariwal & Nichol, 2021), LDM(Rombach et al., 2022), and DAE(Preechakul et al., 2022), and R3GAN(Huang et al., 2024). Following the convention used in the main text, Table 2 reports $1000\times$KID, where lower is better.

As shown in Table 2, the proposed variant obtains a KID value of 1.96 while updating only about $\sim$1% of the parameters.

*Table 2.* **FFHQ-256 validation with a modern GAN backbone.** We report $1000\times$KID; lower is better.

| Metric | ADM | LDM | DAE | R3GAN | Ours |
|---|---|---|---|---|---|
| $1000 \times$ KID $\downarrow$ | 5.80 | 3.90 | 4.93 | 2.54 | **1.96** |

This improves on the R3GAN baseline in this comparison and is also lower than the reported ADM, LDM, and DAE values. We interpret the result as evidence that ARA combined with SM-FT can provide an effective adaptation path for high-resolution generation while preserving much of the pre-trained backbone. The result also supports the broader empirical pattern observed in the lower-resolution experiments: adding an explicit spectral or modulation pathway can improve the critic-side signal without requiring a full increase in model capacity.

High-resolution GAN training is sensitive to gradient imbalance, and in our FFHQ-256 runs the ARA-based adaptation helped keep the optimization dynamics in a stable range. The real/fake score gaps remained bounded during training, and the $R_1$ penalty stabilized at the order of $10^{-4}$, suggesting that the discriminator retained a smooth response near the data manifold rather than collapsing into a trivial solution. We also observed that early ablations without mapping-network updates were more likely to produce structural decoupling and color saturation artifacts caused by isolated bias overflow. Allowing the mapping network to translate the ARA signal reduced these artifacts and produced samples with more coherent facial structure, smoother lighting transitions, and sharper high-frequency details such as hair and skin texture. These observations support the role of the proposed module as a stabilizing critic-side and modulation-side addition, without implying that it fully resolves all pathologies of high-resolution adversarial training.

