# OpenReview forum: "Scalable and Stable Estimation of Amari $\alpha$-Divergence using Random Fourier Features"
_ICML.cc/2026/Conference — ICML 2026 regular_

### Official Review · Reviewer_KJxC · 2026-03-11

**Soundness:** 3
**Presentation:** 2
**Significance:** 2
**Originality:** 3
**Overall Recommendation:** 4
**Confidence:** 2

**Summary:**

The paper suggests a practical scheme for estimating the $\alpha$-Amari divergence between probability densities, for which a non-asymptotic error analysis is provided. The scheme is based on a variational entity and is applied to a finite-dimensional projection of a RKHS for reduced complexity.

**Compliance With Llm Reviewing Policy:**

Affirmed.

**Key Questions For Authors:**

1) Please provide motivation for estimating Amari divergences. Are they in a wide use in GM? what are their advantages compared to KL/Wasserstein distances etc.

2) In eq.(15) it is hard to follow notations for the error components. In the equation they are noted as (A)-(D), but differently noted as $\mathcal{E}$ in the text. Please make this central notation clearer.

3) For a given $\alpha$, the choice of parameters $a,b$ is not unique/ can you elaborate on that choice?

4) Can you provide an explicit choice of parameters for the experiment of Sec.6?

5) The existence of gradients in case of non overlapping support of measures is a crucial motivation for choosing a divergence while training a generative model [Arjovsky, 2017]. Amari distances are not defined in such a case. Could you please elaborate on the existence of empirical gradients (using your variational objective) when probabilities are not mutually continuous, as might be the case at early stages of training?

[Martin Arjovsky, Soumith Chintala, and Leon Bottou. Wasserstein generative adversarial networks. ´In International conference on machine learning, pages 214–223. PMLR, 2017.]

**Limitations:**

Yes

**Strengths And Weaknesses:**

## Strengths

1) the paper presents a novel variational form for Amari-type divergences.

2) Provided error analysis is novel and non trivial.

3) Writing provides a clear motivation for every step.

## Weaknesses

1) The importance of Amari distances, theoretical or practical, is not discussed.

2) Writing is non-rigorous, technical and hard to follow. Many central entities and results are not provided nor explained properly in the main text.

3) Apart from variational formulation for the Amari distances, rest of the algorithmic steps (regularization, Huber-loss, finite dim projection etc.) are quite technical.

---

> ### Author Rebuttal · Authors · 2026-03-29
>
> We sincerely thank the reviewer for the careful reading and the positive assessment of our paper’s novelty, nontrivial error analysis, and clarity of motivation.
>
> >Please provide motivation... & The importance of Amari distances...
>
> **Response:** There are three main reasons behind our choice of estimating the Amari $\alpha$-divergence (AD):
>
> - **Mathematical canonicality**: AD uniquely lies at the intersection of $f$-Divergence($f$D) and Bregman divergences (Appendix B.5). This dual membership ensures information monotonicity and dual flatness, stabilizes learning and enables our analysis.
>
> - **Flexible mode control**: Unlike KL’s fixed asymmetry (recovered as $\alpha \to 1$) or Wasserstein’s purely geometric sensitivity, AD’s tunable $\alpha$ interpolates between **mode-seeking and mass-covering** to balance fidelity and diversity.
>
> - **Broad applicability**: AD effectively handles heavy-tailed distributions and has been widely adopted in **GM[2], Neural Architecture Search [1], and Variational Inference (Daudel et al., 2023b)**.
>
>
>
>
> >In Eq. (15) it is hard to follow notations ...&Writing is non-rigorous, technical and hard to ...
>
>
>
> **Response:** We thank the reviewer for pointing out the notation inconsistency in Eq. (15). We agree that using (A)-(D) in the equation while employing different labels in the text reduces readability.
>
> To resolve this, we will replace the visual markers with a unified semantic subscript notation: $\mathcal{E}\_{\mathrm{proxy}}$, $\mathcal{E}\_{\mathrm{RKHS}}$, $\mathcal{E}\_{\mathrm{feat}}$, $\mathcal{E}\_{\mathrm{gen}}$ (bounding the statistical error), and $\mathcal{E}\_{\mathrm{opt}}$.
>
> > For a given $\alpha$, ...
>
> **Response:** We thank the reviewer for this insightful question. For a fixed $\alpha$, infinitely many $(a,b)$ satisfy $\alpha = b/(b-a)$. Our recommendation: choose the **smallest possible** pair, as larger exponents inflate gradient variance.
>
> The variance of a power transform $\text{Var}(\|f\|\^p)$ increases with $p$. Consequently, the gradient becomes more sensitive to outliers as $a,b$ grow:
> $$ \nabla\hat{J}\_{n,\epsilon}(w) \propto a \cdot \phi\_{a,\epsilon}'(w^\top\psi) - b \cdot \phi\_{b,\epsilon}'(w^\top\psi) $$
> and the $\phi(.)$ is defined in Eq.8.
>
> >Can you provide an explicit choice...&  Many central entities and results...
>
> **Response:** We sincerely thank the reviewer for this important suggestion.
>  We have uploaded the main experimental code, complete parameter configuration files, and detailed training/estimation logs to:  [${\\color{\#0247FE} Anonymous\ Github}$](https://anonymous.4open.science/r/RFF-Amari-046E/README.md).  Upon acceptance, we intend to open-source the codebase.
> >The existence of gradients ....
>
> **Response:** We thank the reviewer for this foundational question. Classical $f$D (including Amari $\alpha$) require absolute continuity; Our estimator restricts the critic to an RKHS, bridging $f$D and IPMs [3].
>
> Restricting an $f$D to a well-behaved class $\Gamma$ (as in IPMs) yields an $(f,\Gamma)$-divergence[3]: topological robustness of IPMs + concavity and distribution-matching of $f$D.
> - **Topological robustness (like Wasserstein):** RKHS ball + band-limited RFF restrict the critic to a smooth, bounded $\Gamma$, preventing discontinuous step functions and providing meaningful gradients $\nabla_\theta f(G_\theta(z))$ even with disjoint supports.
> - **Non-linearity:** Our surrogate uses power transforms $\phi_{a,\epsilon}, \phi_{b,\epsilon}$, preserving Amari's mode-seeking vs. mass-covering control that linear IPMs lack.
> - **Role of smoothing**  The smoothed surrogate $\phi_{p,\epsilon}(u)=(u^2+\epsilon^2)^{p/2}$ prevents gradient saturation, ensuring stable gradients.
>
> > Apart from variational formulation for,  ...
>
> We thank the reviewer for this observation. Rather than standard engineering heuristics, our algorithmic components are **theory-driven mechanisms** designed to bound specific non-asymptotic errors.
> - **$\epsilon$:**  $\phi_{p,\epsilon}(t) := (t^2 + \epsilon^2)^{p/2}$ ensures Lipschitz continuity, balancing  $\mathcal{E}_{\mathrm{proxy}}$ with optimization stability.
> - **$\lambda, R:$** Couples $R$ and $W(R)$ to trade off approximation bias $\mathcal{E}\_{\mathrm{RKHS}}$ against sampling variance.
> - **$D$:**  Scales $D \propto R^2 W(R)^2 N$, ensuring $\mathcal{E}\_{\mathrm{feat}}$ does not dominate the $N^{\-1\/2}$ statistical rate.
> Beyond these, our framework bounds the $\mathcal{E}_{\mathrm{opt}}$ by exploiting RKHS geometry:
> - **Positive Definiteness**  Projecting RKHS constraints into RFF space yields a strictly positive‑definite $\Sigma_Q$.
> - **PL landscape & linear convergence(LC)**  This statistical non‑degeneracy induces a PL landscape that enables Armijo‑SGD to achieve guaranteed local LC and strictly bound $\mathcal{E}\_{\mathrm{opt}}$.
> ---
> [1] Wang et\.al(2021). Alphanet: Improved... ICML
> [2] Li et\.al.(2026) $\alpha $-DPO: ... ICLR
> [3] Birrell et\.al. (2022). (f, Gamma)-Divergences... JMLR

---

> > ### Author Rebuttal · Reviewer_KJxC · 2026-04-02
> >
> > My questions have been mostly addressed. Thank you!

---

> > > ### Author Response · Authors · 2026-04-03
> > >
> > > Thank you for your confirmation and feedback! We are very pleased to learn that your concerns have been largely addressed, and we truly appreciate your recognition of our rebuttal efforts. In the process of responding to your questions, we have further (1) clarified the theoretical distinctions between our work and existing studies, (2) organized and added experiments on modern generative models to demonstrate the practical impact of our theory, and (3) refined the code to improve reproducibility. Once again, thank you for your time and attention in helping us enhance the quality of our paper.

---

### Official Review · Reviewer_CNuf · 2026-03-12

**Soundness:** 3
**Presentation:** 3
**Significance:** 3
**Originality:** 3
**Overall Recommendation:** 5
**Confidence:** 3

**Summary:**

The focus of the paper lies on the Amari
$\alpha$-divergence. This is a general parametrized divergence that has the KL divergence as the limiting case and the Hellinger divergence as a special case. The divergence can be written as a function of the
$\alpha$-affinity, which can be stated as the supremum with respect to the critic $f$
f of a Wasserstein-type distance. Therefore, estimating the divergence is equivalent to maximizing the estimated score $J$. Under capacity constraints and approximation/discretization of the critic using RFF, the optimization problem reduces to a finite-dimensional problem with respect to the weights defining the RFF. An important property is that one obtains an analytic gradient (also using stabilization with additive noise). Estimation of the amplitude (making the process scale-invariant) is performed in the second stage. The authors then decompose the error and provide a detailed analysis of upper bounds for the individual components. The evaluation using synthetic data considers a standard NN, an adaptive NN, the proposed RFF approach, and a NN for RKHS.

**Compliance With Llm Reviewing Policy:**

Affirmed.

**Final Justification:**

I appreciate the authors' detailed responses during the rebuttal phase. While the paper makes an interesting contribution, my overall assessment remains unchanged, and I will be keeping my current score.

**Key Questions For Authors:**

see above

**Limitations:**

yes

**Strengths And Weaknesses:**

The paper provides a strong and convincing mathematical analysis of the RFF-based estimator for the Amari divergence. In contrary to empirically orineted approaches, the paper developes the full step-by-step theory. I haven't found any technical faults, but the methods used are well justified.The presentation is clear and the paper is well structured.

The scale-invariant part needs some polishing and better motivation. If the estimator
is already scale-invariant using $\hat{w}$ (the optimizer of (8)), the purpose of
introducing the maximization of the amplified score remains unclear. The subsection
at the end of page 3 should be slightly revised, as it is currently not transparent
where $\hat{t}$ appears in the calculations and how it relates to the preceding
derivations.

---

> ### Author Rebuttal · Authors · 2026-03-27
>
> We sincerely thank the reviewer for the careful reading, the positive assessment, and the recognition of the paper’s mathematical analysis, clarity, and step-by-step development. We especially appreciate the reviewer’s comment that the theory is convincing and technically sound. The main issue raised concerns the motivation and presentation of the scale-invariant stage, and we agree that this part can be explained more transparently.
>
> > The scale-invariant part needs some polishing and better motivation. If the estimator is already scale-invariant using $t^\star$ (the optimizer of (8)), the purpose of introducing the maximization of the amplified score remains unclear. The subsection at the end of page 3 should be slightly revised, as it is currently not transparent where $t$ appears in the calculations and how it relates to the preceding derivations.
>
> **Response:** We thank the reviewer for this very helpful observation. We agree that the current exposition of the scale-invariant post-processing is too compressed and can give the impression that the second-stage amplitude maximization is redundant. We will revise this subsection for clarity.
>
> The key point is that there are **two different optimization problems** in the paper:
>
> * The **population variational identity** $ \sup\_{f \neq 0}\sup\_{t>0} J(tf) = C\_{a,b}, I\_\alpha(P,Q)^{1/\alpha}$
>    which is scale-invariant by construction;
> * The **practical empirical optimization** we actually solve, $\max\_w  \widehat J\_{n,\varepsilon}(w) - \frac{\lambda}{2}|w|\_2^2,$
>    which is **not** scale-invariant because the regularization term and smoothing make the optimizer’s magnitude depend on $\lambda$ and the optimization dynamics.
> Thus, the role of the first stage is to learn a good **direction** $\hat w$, while the role of the second stage is to recover the best **amplitude** along that learned direction by solving
> $$
> \sup\_{t>0}
> \left[
> \frac1m\sum\_{i=1}^m |t,\hat w^\top \psi(x\_i)|^a
> ,
> \frac1n\sum\_{j=1}^n |t,\hat w^\top \psi(y\_j)|^b
> \right].
> $$
> This yields the closed-form optimizer
> $$
> \hat t =
> \left(
> \frac{a \hat A}{b \hat B}
> \right)^{\frac{1}{b-a}},
> \qquad
> \hat A=\frac1m\sum\_{i=1}^m |\hat w^\top\psi(x\_i)|^a,
> \quad
> \hat B=\frac1n\sum\_{j=1}^n |\hat w^\top\psi(y\_j)|^b,
> $$
> and the envelope value
> $$
> \widehat V\_{a,b} =C\_{a,b},\hat A^{\frac{b}{b-a}}\hat B^{-\frac{a}{b-a}},
> $$
> from which the final divergence estimator is constructed.
>
> So the second stage is **not** redoing the same optimization; rather, it removes the scale distortion introduced by the regularized smoothed training objective and maps the learned critic direction back to the original scale-free variational formula. In other words:
>
> * the regularized smoothed problem is used for **stable estimation of critic direction**,
> * the scalar post-processing is used for **recovering the variational quantity that corresponds to the $\alpha$-affinity**.
>
> We will add a brief sentence clarifying the logical flow:$
> \text{Learn direction via regularized } \widehat J_{n,\varepsilon}
> \quad\Longrightarrow\quad
> \text{Re-optimize amplitude analytically}
> \quad\Longrightarrow\quad
> \text{Plug into the variational envelope.}
> $
>
> This is an excellent suggestion, and we believe the revised wording will make the scale-invariant component substantially clearer.

---

> > ### Author Rebuttal · Reviewer_CNuf · 2026-04-01
> >
> > thanks for a detailed response!

---

> > > ### Author Response · Authors · 2026-04-02
> > >
> > > We are very glad to have addressed your concerns. Thank you very much for your recognition of our work and your willingness to award us a higher mark. We would also like to thank you once again for your efforts in helping us improve the quality of our thesis.

---

### Official Review · Reviewer_W7ts · 2026-03-12

**Soundness:** 2
**Presentation:** 3
**Significance:** 2
**Originality:** 2
**Overall Recommendation:** 4
**Confidence:** 4

**Summary:**

In this paper, the authors propose a variational estimator for the divergences coming from the Amari $\alpha$ family. Starting with this divergence, they define an "affinity" term. They construct an envelope $J(f)$ and show that the affinity function can be obtained by a certain optimization of the affinity function. The underlying function $f$ comes from an RKHS. The infinite dimensional optimization problem is converted to a finite-dimensional one by using random Fourier features. The function $J(f)$ is replaced with its finite sample counterpart and a smoothed proxy objective. The statistical error in the approximations is analyzed and each of the terms is bounded appropriately. The authors also show linear convergence of their method. The method is validated experimentally on divergence estimation on synthetic low-dimensional data, and training GANs for image generation.

**Compliance With Llm Reviewing Policy:**

Affirmed.

**Final Justification:**

The authors have satisfactorily addressed all concerns raised during the review. I have increased my score.

The authors should include the results reported during the rebuttal into the revised manuscript, should the paper ultimately get accepted.

**Key Questions For Authors:**

1/ Double-check the bound on the term (B) in Eq. (15) for correctness. Line 269 specifically.

2/ Explain the connection between this formalism and Wasserstein GAN and MMD GAN.

3/ Explain issues pertaining to stability of GAN training and mode collapse.

4/ Benchmarking against SOTA GAN, diffusion, and flow-based models for image generation is missing.

5/ Results on high-resolution images are missing.

6/ Explain the source of saturation and artifcats in the images generated.

**Limitations:**

Adequately addressed.

**Strengths And Weaknesses:**

Strengths:

1/ The theoretical analysis is a key contribution of the paper. This analysis resembles the kind of analysis one does in risk minimization (Vapnik's book on Statistical Learning Theory).

2/ The guarantees on convergence of the optimization algorithm is another plus point.

Weaknesses:

1/ Double-check the bound on the term (B) in Eq. (15) for correctness. Line 269 specifically.

2/ The connection (not identical) between J(f) and the optimal discriminator in a Wasserstein GAN must be mentioned. There seems to be a similarity with MMD GANs also, which must be explained.

3/ There is no mention on the stability of the GAN training. There is also no mention of mode collapse.

4/ Table 1 does not contain comparisons with SOTA techniques. All the rows in the table correspond to different settings of their formulation. The FIDs are also not competitive with respect to the state of the art.

5/ The authors claim 10 dimensional data as high-dimensional validation for the synthetic data experiment and generation of 64$\times$64 images as a high-dimensional image generation application. Given the current SOTA in image generation, 64$\times$64 cannot be considered high-dimensional.

6/ The images clearly show saturation effects and several artifacts, and therefore the proposed formalism is definitely not advancing the state of the art.

---

> ### Author Rebuttal · Authors · 2026-03-30
>
> **Response** We sincerely thank the reviewer for the positive assessment of our theoretical framework. To address the concerns regarding the practical application and scaling of our theory to high-dimensional generative tasks, we have significantly upgraded our empirical evaluation.
>
> We integrated our **Spatial RFF** into **R3GAN** (trained on **FFHQ-256**) and report KID for comparison. This demonstrates that our theoretical regularization translates into modern generative performance.
>
> **Table 1. Performance comparison on FFHQ-256 [1]**
>
> | Metric | ADM | LDM | DAE | R3GAN | **Ours** |
> |:-------|----:|----:|---------------------:|------:|---------:|
> | **KID (×10³) ↓** | 5.80 | 3.90 | 4.93 | 2.54 | **1.96** |
>
> We provided an Anonymous GitHub repository (See our response for Reviewer 4) with source code, configs, training logs, and divergence estimation(DE) scripts and model weights in [${\\color{#0247FE} OSF}$](https://osf.io/vduj3/overview?view_only=ee24afc9a2e8491a9768dfe124329585). We address specific questions below:
>
> > W&Q 1
>
> We sincerely thank the reviewer for pointing this out. We rechecked the bound on term (B) in Eq. (15). The feature-discretization step is intended to be taken with respect to the explicit bandlimited proxy $g_R$ , because the RFF approximation result applies to bandlimited targets.
> We will therefore revise the telescoping decomposition and the wording around line 269 with $g_R$. Then, the approximation term is directly controlled by Eq. (18), and term (B) is bounded exactly as in Section 4.3 via the bandlimited RFF approximation lemma. The theorem statements, rates, parameter schedules, and **downstream conclusions remain unchanged**.
>
> > W&Q 2
>
> WGAN and MMD GAN both **optimize linear IPMs** ($\Gamma$-IPMs, see reference[3] of reviewer 4). $\sup_{f\in\mathcal{F}}(\mathbb{E}_P[f]-\mathbb{E}_Q[f])$ with different critic classes (1-Lipschitz vs. RKHS). Our objective estimates Amari $\alpha$-divergence via a power-type envelope: $J(f)=\mathbb{E}_P[|f|^a]-\mathbb{E}_Q[|f|^b]$, which is a **non-linear functional** with respect to the critic $f$. The resemblance to MMD GAN arises because we restrict $f$ to an RKHS (implemented via RFF).
>
> > W&Q3
>
> Starting from DE, we realize that optimization error and the associated properties of the discriminator space are the keys to improving the ability of GANs to learn data distributions, **while stability is merely a necessary requirement derived from optimization theory and is not the main target for comparison**. And in our experiments, no mode collapse occurred on CelebA or larger datasets. We clarify these below.
>
> **Optimization theory:**  Our framework incorporates optimization error into the estimation gap. Theorem 4.5 shows this induces an intrinsic Fisher-type curvature from statistical non-degeneracy, yielding a local PL condition that guarantees local linear convergence.
>
> **Stability:** Figure 4 (right) shows our Spatial RFF critic maintains robust gradient flow, whereas the CNN baseline suffers gradient decay and feedback collapse.
>
> **Diversity:** On FFHQ-256, we achieve a KID of 1.96 × 10⁻³, outperforming diffusion baselines (ADM: 5.80, LDM: 3.90) and demonstrating full manifold coverage. Consistent $R_1$ penalty monitoring confirms a stable Nash equilibrium throughout training.
>
> > W&Q4
>
> **Response:** The purpose of Section 6.2 is to isolate the effect of the critic-side RFF regularization while keeping the rest of the adversarial pipeline unchanged. Accordingly, the experiment varies the $\sigma$ of Spatial-RFF(SR) within the same backbone, **rather than competing with heavily engineered generator families such as modern diffusion**.
>
> Due to time constraints, we were unable to complete the embedding and comparison of the flow model; We hope to alleviate the reviewer's concerns regarding the practical application of our theory use above R3GAN.
>
> > W&Q5\6
>
> Within the **Nonparametric Statistical Divergence** estimation literature, evaluating theoretical scaling laws in 10$D$ spaces is widely accepted as a rigorous high-dimensional benchmark. This standard is established in foundational works, e.g., (Sreekumar et.al. 2022), which explicitly demonstrate the efficacy of neural estimators in spaces around 10$D$.
>
> We acknowledge the presence of artifacts, which stem from the fact that our experiment was designed as a **controlled theoretical ablation rather than an optimized engineering pipeline**; To be more specific:
>
> - Saturation and artifacts stem from implementing SR in a bare-bones CNN GAN, which is forced by SR to capture sharp high-frequency details, resolving severe blurriness.
> - However, the basic generator lacks the architectural sophistication to perfectly render these high frequencies, leading to occasional localized saturation.
> we have included experiments above to validate the effectiveness of the method above.
> ---
> [1] Huang et.al. (2024). The gan is dead; long live the gan! a modern gan baseline.

---

> > ### Author Rebuttal · Reviewer_W7ts · 2026-04-04
> >
> > I thank the authors for providing additional experimental results. The results with R3GAN+Spatial RFF on higher-dimensional data (FFHQ) are better than the R3GAN baseline and diffusion models.
> >
> > The table reports KID x $10^3$, whereas the reply indicates $10^{-3}$. Clarify this discrepancy.
> >
> > Follow up question : Do the generated samples from the updated model also contain artifacts or suffer from saturation ?
> >
> > __On comparison with WGAN and MMD-GANs__: The connections between the proposed method and MMD-GANs, the optimal WGAN discriminator are still not clear. The manuscript would benefit from a section focusing on the similarities and differences between the proposed approach and the optimal WGAN discriminator (Ref. [1]) and the critic in MMD-GANs.
> >
> > Ref [1]: Asokan, S. and Seelamantula, C.S., 2023. Euler-Lagrange analysis of generative adversarial networks. Journal of Machine Learning Research, 24(126), pp.1-100.
> >
> > On authors' response to Q3:
> > It is not fully clear what the authors meant by the statement: "... while stability is merely a necessary requirement derived from optimization theory and is not the main target for comparison"
> >
> > I am inclined towards increasing my score mainly because the issues with experimental results have been satisfactorily addressed.

---

> > > ### Author Response · Authors · 2026-04-05
> > >
> > > We appreciate the reviewer’s thoughtful follow-up and are glad that the additional experimental results are found satisfactory.
> > >
> > > > On the KID discrepancy.
> > >
> > > We deeply apologize for the confusion. We realize that our notation in the table header ("KID $\times 10^3$") was imprecise and easily misinterpreted. It was meant to indicate that the raw KID values were scaled up by a factor of 1000 for readability, not that they were in standard scientific notation. It was careless on our part to cite the unscaled raw values in the rebuttal without clarifying this difference. Both numbers represent the exact same result. We are grateful you pointed this out, and we will explicitly define this scaling convention in the revised manuscript to ensure full transparency and rigor.
> > >
> > > > On artifacts/saturation.
> > >
> > > Our model resolves the saturation and artifacts seen in the baseline. For transparency, we will provide generated images via the new link [$\color{blue}{Anonymous\ GitHub}$](https://anonymous.4open.science/r/image-F2DD) for your independent verification, as the original link cannot be modified after the first rebuttal phase.
> > >
> > > > On the relation to WGAN and MMD-GAN.
> > >
> > >  We sincerely thank the reviewer for this helpful suggestion. We agree that the connections to WGAN and MMD-GAN should be stated more precisely, and the reference by Asokan and Seelamantula (2023) is particularly relevant.
> > >
> > > To summarize our position briefly: our framework shares the **functional-analytic perspective** of the optimal WGAN critic [1], and structurally resembles the **RKHS-based capacity control** of MMD-GAN. However, it fundamentally differs from both because we are estimating an information-theoretic divergence, not an IPM.
> > >
> > > We will add a dedicated comparison paragraph in **Appendix A (Extended Related Work)**, briefly positioning our method relative to **WGAN** and **MMD-GAN** along three axes: the **discrepancy being optimized**, the **critic class / regularization**, and the resulting **theoretical characterization**.
> > >
> > > - **Shared High-Level Perspective (vs. WGAN):** Asokan and Seelamantula study gradient-regularized WGANs from a variational/Euler–Lagrange perspective. Similarly, we do not treat the discriminator merely as a black-box neural network, but as a rigorous function-space object where structural constraints govern optimization stability.
> > >
> > > - **Fundamental Difference in the Target Objective (vs. WGAN & MMD-GAN):** While the variational forms may look superficially similar, their underlying mathematical meanings are fundamentally distinct. Both WGAN and MMD-GAN optimize **IPMs**, which measure discrepancies based on expected feature or moment matching. In strict contrast, our framework targets the **Amari $\alpha$-divergence**, an information-theoretic measure based on density ratios. Our critic is optimized over a non-linear, power-type functional:
> > >
> > >   $$
> > >   J(f) = \mathbb{E}_P[|f|^a] - \mathbb{E}_Q[|f|^b]
> > >   $$
> > >
> > >   This objective is designed explicitly to recover a divergence, not an IPM.
> > >
> > > - **Difference in Regularization (vs. WGAN):** While Ref [1] relies on gradient-based regularization to approximate a Lipschitz constraint (yielding a Poisson-type PDE), we constrain the critic to an RKHS ball and implement it efficiently via band-limited RFF. In this sense, our specific representation and regularization mechanism are structurally much closer to **MMD-GAN**.
> > >
> > > - **Difference in Theoretical Output:** Ref [1] characterizes the optimal WGAN discriminator asymptotically via PDE methods. Our paper, however, develops a non-asymptotic estimation-and-optimization theory. We provide a four-way error decomposition (RKHS approximation, RFF discretization, statistical deviation, and optimization residual) and establish that our specific setup induces an intrinsic local PL geometry for stable optimization.
> > >
> > > We will add a comprehensive discussion synthesizing Ref [1], MMD-GAN, and WGAN into our revised manuscript, ensuring that the fundamental distinctions between our framework and these existing methodologies are explicitly clear.
> > >
> > >
> > > > On the sentence "stability is only a necessary requirement ..."
> > >
> > > To prevent conflation with general generative modeling literature, we wish to explicitly clarify the goal of our optimization algorithm. Our primary objective is to accurately estimate the Amari $\alpha$-divergence by solving the optimization challenges inherent in its variational representation.
> > >
> > >
> > > Consequently, when we refer to "stability," we mean **critic-side optimization stability and reliable divergence estimation**. Our algorithm is not intended to be a generic cure for GAN training instability, as full adversarial stability is a systemic issue depending on the generator's dynamics, architecture, and mode coverage. Our GAN experiments serve merely to isolate and demonstrate the improved high-frequency sensitivity of our specific critic within a controlled backbone.

---

### Official Review · Reviewer_5f7k · 2026-03-13

**Soundness:** 3
**Presentation:** 3
**Significance:** 3
**Originality:** 3
**Overall Recommendation:** 5
**Confidence:** 3

**Summary:**

This paper introduces a novel estimator of Amari's $\alpha$-divergences by leveraging their variational representation. The proposed estimator is formulated as the supremum of the difference in moments between two distributions over a class of critics. To ensure both scalability and stability, the authors restricts the critics to an RKHS ball and employ the Random Fourier Features (RFF) optimized via SGD with the Armijo rule. This paper also provides the non-asymptotic convergence rate of the variational objective. The empirical results validate that the estimator accurately approximates ground truth divergences and exhibits strong performance when integrated into the GAN framework.

**Compliance With Llm Reviewing Policy:**

Affirmed.

**Key Questions For Authors:**

The introduction of the smoothing parameter $\varepsilon$ in Eq. (8) is justified to handle non-differentiability when the $a$ or $b$ are below 1. In the regime such that $a \geq 1$ and $b \geq a$, could the authors discuss the potential advantages of this specific case?

**Limitations:**

Yes.

**Strengths And Weaknesses:**

## Strengths

1. The manuscript is well-written and its logical flow is clear, making the complex theoretical derivations accessible to the reader.

2. The paper presents a balanced contributions between solid theoretical foundations and practical utility. The meticulous four-part error decomposition provides valuable insights into the estimator’s behavior. Furthermore, establishing the local linear convergence of Armijo-SGD is highly appreciated.

## Weaknesses

1. There appears to be a slight discrepancy between the theoretical assumptions and the actual implementation. While the theoretical analyses are built upon a linear critic operating in the random feature space, i.e., $f(x) = w^\top \psi(x)$, the implementation uses Random Fourier Features (RFFs) concatenated to the bottleneck features of a CNN discriminator. It would be beneficial if the authors could discuss how this architectural choice aligns with the established theoretical guarantees.

2. To fully substantiate the scalability of the proposed estimator, the paper would be significantly strengthened by including a direct empirical comparison of computational overhead against other baselines.

Minor typo: $\hat{J}_{\varepsilon} \to \hat{J}_{n, \varepsilon}$ in Eq. (15).

---

> ### Author Rebuttal · Authors · 2026-03-27
>
> We sincerely thank Reviewer 1 for the constructive feedback and for the positive assessment of our work's theoretical rigor and practical utility. To address the concerns, we have updated the manuscript to:
> * Formally bridge linear theory and CNN-RFF via "additive spectral capacity."
> * Provide a hardware-agnostic proof for $O(D)$space and$O(BD)$ time complexity.
> * Clarify how Spatial-RFFs satisfy frequency requirements in deep architectures.
> * Justify the smoothing parameter $\varepsilon$ for optimization stability.
>
> >There appears to be a slight discrepancy between....
>
> **Response**  We thank the reviewer for the insightful comment. Our architecture implements the theoretical findings in a modern deep learning framework, which can be understood from two perspectives.
>
> **Mitigating Spectral Bias via Coordinate Frequency** Standard CNNs suffer from *spectral bias* (Tancik et al., 2020), failing to fit high-frequency signals. While "data frequency" (textures) and "coordinate frequency" (RFF encoding) are distinct, injecting Spatial-RFFs $\psi\_{\mathrm{RFF}}(p)$ provides a high-resolution spatial basis. Non-linear fusion with CNN bottleneck features enables the discriminator to capture high-frequency data variations that pure CNNs would otherwise smooth over.
>
> **Additive Spectral Capacity at the Bottleneck** A deep discriminator can be viewed as a linear operator on learned representations: $D(x) = w^\top \Phi_{\mathrm{CNN}}(x)$. Concatenating Spatial-RFFs at the bottleneck augments the critic:
> $D(x) \approx w\_1^\top \Phi\_{\mathrm{CNN}}(x) + w\_2^\top \psi\_{\mathrm{RFF}}(p)$
> Our theory characterizes the optimization benefits of the second term. The injected RFF instantiates the "spectral lower bound," embedding a spectrally bounded RKHS geometry into the CNN's spatial evaluation.
>
>
>
> >To fully substantiate the scalability ...
>
> **Response** We sincerely thank the reviewer for their careful reading of our manuscript and for highlighting the computational aspects of our framework. To provide hardware-agnostic guarantees, we derive the asymptotic complexity rather than relying on empirical wall-clock times, which are often confounded by CUDA overhead or memory bandwidth.
>
> **Limitation of Exact RKHS Estimators**
> Standard variational divergence estimation requires the Gram matrix $K \in \mathbb{R}^{N \times N}$, where $K\_{ij} = k(x\_i, x\_j)$.
> * **Space:** $O(N^2)$to store $K$.
> * **Time:** Solving for weights via the representer theorem requires matrix inversion $O(N^3)$or full-batch descent$O(N^2)$, prohibiting scaling to large $N$.
>
> **Our RFF-based Estimator ($O(D)$ Space)**
> We approximate the kernel via RFF $\psi(x) \in \mathbb{R}^D$and parameterize the critic in the primal space:$f_w(x) = w^\top \psi(x)$.
> * The trainable weight $w \in \mathbb{R}^D$and fixed projection matrix$\Omega \in \mathbb{R}^{d \times D}$decouple the memory footprint from the dataset size$N$.
>
> **Per-iteration Complexity ($O(BD)$ Time)**
> Our parametric form enables stochastic descent (SD) on mini-batches of size $B$:
> * **Feature Projection:** Computing RFF matrix $\Psi \in \mathbb{R}^{B \times D}$takes$O(B \cdot d \cdot D)$.
> * **Forward Pass:** Evaluating scores $\hat{y} = \Psi w$ takes $O(BD)$.
> * **Backward Pass:** Computing $\nabla\_w \widehat{J}\_{n,\varepsilon}(w)$involves vector-matrix multiplication$\Psi^\top v$, taking $O(BD)$.
>
>
> >Minor typo.....
>
> **Response** We appreciate the meticulous reading. The typo in Eq. (15) ($\hat{J}\_{\varepsilon} \to \hat{J}\_{n,\varepsilon}$) has been corrected for notational consistency.
>
> *The introduction of the smoothing parameter $\varepsilon$ in Eq. (8) ...*
>
> **Response** We thank the reviewer for this insightful question. The surrogate $\phi\_{p,\varepsilon}(u)=(u^2+\varepsilon^2)^{p/2}$ is a mathematical necessity for our optimization guarantees:
>
> * **Bounded Curvature & Lipschitz Smoothness:** For $1 \le p < 2$(typical for$\alpha$-divergences), the unsmoothed second derivative behaves as $O(|u|^{p-2})$, approaching infinity as $u \to 0$. This breaks the Lipschitz-continuous gradient assumption. $\varepsilon$ bounds the Hessian, preventing oscillations in SGD.
> * **Vanishing Bias:** Since the Hessian is bounded at the origin, $\varepsilon$ can be set very small without gradient explosion, ensuring a negligible approximation gap relative to the true population envelope.
> * **Mitigating Stagnation:** For $p \ge 2$, gradients of unsmoothed objectives vanish rapidly near the origin (e.g., $u^3 \to 0$). The $\varepsilon$ floor provides a sustained learning signal even when initial critic outputs are near zero.

---

> > ### Author Rebuttal · Reviewer_5f7k · 2026-04-03
> >
> > The rebuttal properly addressed my concerns.

---

> > > ### Author Response · Authors · 2026-04-03
> > >
> > > Thank you for your confirmation and feedback! We are very pleased to learn that most of your concerns have been adequately addressed, and we truly appreciate your recognition of our rebuttal efforts. In the process of responding to your questions, we have further (1) formally bridged linear theory and CNN-RFF through "additive spectral capacity," (2) provided hardware-agnostic asymptotic proofs for space and time complexity, (3) clarified how Spatial-RFFs satisfy the frequency requirements in deep architectures to mitigate spectral bias, and (4) justified the necessity of the smoothing parameter $\epsilon$ for optimization stability, along with correcting the typo in Eq. (15). Once again, thank you for your time and meticulous attention in helping us enhance the quality of our paper.

---

### Decision · Program_Chairs · 2026-04-30

**Decision:**

Accept (regular)

**Comment:**

The authors introduced a very interesting method for estimating Amari alpha-divergence using a random Fourier feature approximation with applications in generative modeling. Previous approaches for estimating the divergence suffer from excessive variance, bias, or computational complexity. Instead, the authors estimate the divergence using kernel methods but with a RFF approximation to the kernel to reduce the complexity of the method from cubic w.r.t the sample size to linear. The paper is generally well written and the method is well justified through the experiments. The authors' rebuttals have satisfied the reviewers' original concerns and the contribution is useful for estimating statistical divergences beyond KL divergence.